

# ENSO - cave dripwater hydrochemical relationship: a 7-year dataset from SE Australia

Carol V. Tadros[1,2], Pauline C. Treble[1,2], Andy Baker[2], Ian Fairchild[3,4], Stuart Hankin[1], Regina Roach[5], Monika Markowska[1,2], Janece McDonald[6]

[1]Australian Nuclear Science and Technology Organisation, Locked Bag 2001, Kirrawee DC, NSW 2232, Australia
[2]Connected Waters Initiative Research Centre, UNSW Australia, Sydney, NSW, Australia
[3]School of Geography, Earth and Environmental Sciences, University of Birmingham, Edgbaston, Birmingham, UK
[4]Birmingham Institute for Forest Research, University of Birmingham, Edgbaston, Birmingham, UK
[5]NSW National Parks and Wildlife Service, Sydney, NSW, Australia
[6]Environmental and Climate Change Research Group, School of Environmental and Life Sciences, University of Newcastle, Callaghan, NSW 2308, Australia

*Correspondence to:* Carol V. Tadros (Carol.Tadros@ansto.gov.au)

**Abstract.** Speleothems (cave deposits), used for paleoenvironmental reconstructions, are deposited from cave dripwaters. Differentiating climate and karst processes within a dripwater signal is fundamental for the correct identification of paleoenvironmental proxies and ultimately their interpretation within speleothem records. We investigate the potential use of trace element and stable oxygen-isotope ($\delta^{18}O$) variations in cave dripwaters as paleorainfall proxies in an Australian alpine karst site. This paper presents the first extensive hydrochemical and $\delta^{18}O$ dataset from Yarrangobilly Caves, in the Snowy Mountains, south-east Australia. Using a 7-year long rainfall $\delta^{18}O$ and dripwater Ca, Cl, Mg/Ca, Sr/Ca and $\delta^{18}O$ dataset from three drip sites, we determined that the processes of mixing, dilution, flow path change, carbonate mineral dissolution and Prior Calcite Precipitation (PCP) accounted for the observed variations in the dripwater geochemical composition. We identify that the three monitored drip sites are fed by fracture flow from a well-mixed epikarst storage reservoir; supplied by variable concentrations of dissolved ions from soil and bedrock dissolution. We constrained the influence of multiple processes and controls on dripwater composition in a region dominated by ENSO. During the El Niño and dry periods, enhanced PCP, a flow path change and dissolution due to increased soil $CO_2$ production occurred in response to decreased rainfall, in distinction to the La Niña phase where dilution dominated and reduced PCP was observed. We present a conceptual model, illustrating the key processes impacting the dripwater chemistry. We identified a robust relationship between ENSO and dripwater trace element concentrations and propose variations in speleothem Mg/Ca and Sr/Ca ratios may be interpreted to reflect paleorainfall conditions. These findings inform paleorainfall reconstruction from speleothems regionally and provide a basis for paleoclimate studies globally, in regions where there is intermittent recharge variability.

## 1 Introduction

The El Niño-Southern Oscillation (ENSO) is the leading mode of rainfall variation in SE Australia (Dai et al., 1997) where extreme events of rainfall variability such as droughts, floods, bush fires and cyclones associated with ENSO are prominent





(Risbey et al., 2009). Severe drought between 2001 to 2008 and enhanced El Niño conditions, resulted in record low inflows from the alpine headwaters of the Murray River (Murphy and Timbrel, 2008; Cai and Cowan, 2008; Nicholls, 2010), strongly impacting water resource availability of the Murray-Darling Basin and agricultural production; affecting the livelihood of urban and rural Australians (Barros and Bowden, 2008; McGowen et al., 2009). Reconstructing past ENSO variability, from speleothems (calcium carbonate cave deposits) located at Yarrangobilly Caves in the Snowy Mountains alpine region will provide a basis for understanding future regional impacts, therefore assisting with water resource management policy making and the global impacts that ENSO driven climate variability has on the environment; agricultural production, water resources, ecosystems and on human life; emergency management and disease (Power and Smith, 2007).

Studies have shown that trace element time series constructed from the central growth axis of a speleothem, provide potential proxy evidence of paleorainfall conditions (Roberts et al., 1998; Lauritzen et al., 1999; Fairchild et al., 2001; Johnson et al., 2006; Cruz et al., 2007; Jo et al., 2010). The concentration of trace elements in dripwater is dependent on the evolution of the dripwater geochemistry; which is influenced by site-specific characteristics (Spötl et al., 2005) and a range of surface and karst processes (Baldini et al., 2006). Elements may be atmospherically derived from meteoric precipitation, dust supply (Goede et al., 1998; Dredge et al., 2013), marine aerosols (Baker et al., 2000; Fairchild et al., 2000), volcanic eruption activity (Frisia et al., 2005, 2008), atmospheric pollutants (Spötl et al., 2005; Wynn et al., 2008), the host rock and soil (Tooth and Fairchild, 2003). Surface processes; deforestation (Borsato et al., 2007) and fire (Coleborn et al., 2016; Nagra et al., 2016); soil processes; water/sediment and water/rock interaction (Fairchild et al., 2000), colloid, particle or solute mobilisation (Hartland et al., 2012), and temperature, water availability and $CO_2$ changes (Cuthbert et al., 2014; Rutlidge et al., 2014; Treble et al., 2016); and karst hydrological processes; hydrological flow routes, mixing and dilution effects, degassing and calcite precipitation, differential dissolution and incongruent dissolution and selective leaching (Fairchild et al., 2000; Tooth and Fairchild, 2003), potentially modulate the concentration of elements in the dripwater. Constraining these processes and understanding potential climatic signals in the hydrochemistry of dripwater is quintessential in successfully using trace elements as a paleoclimate proxy (Fairchild et al., 2006).

As such, long term datasets of stable oxygen-isotope ($\delta^{18}O$) and geochemistry (trace element concentrations and ratios) of rainfall and cave dripwater provide an empirical basis for identifying factors influencing trace element variability ultimately recorded in speleothems (Baldini et al., 2002; Treble et al., 2003; Riechelmann et al., 2011; Oster et al., 2012; Frappier, 2013; Partin et al., 2013). For example, McDonald et al. (2004) demonstrated Mg/Ca and Sr/Ca ratios in the dripwater doubled in response to an El Niño event which occurred during a 2.5 year baseline monitoring study at Wombeyan Caves, SE Australia. This was an important finding that raised the potential for using speleothem records to reconstruct past ENSO variability for this region. The Wombeyan Caves site lies in the Sydney catchment. The study site used here lies ~85 km away in the headwaters of the Murray Darling basin and as such provides an opportunity to further examine the ENSO signal in cave dripwaters at a second site from this region, with a longer dataset. Additionally, the interpretation of this new



hydrochemical dataset is conducted within an established framework. The dominant controls on precipitation stable isotope variability in this alpine region have been examined. Callow et al., 2014 conducted event based $\delta^{18}$O precipitation (rainfall and snow) sampling across 18 sites (n=70; from February 2010 to March 2012) from a transect in the Snowy Mountains, they determined the origin of moisture, pathway and terrain effects were the dominant controls on precipitation stable

isotope variability in this alpine region. Also the unsaturated zone hydrology of our studied cave has been investigated. Markowska et al., (2015) presented rainfall, soil moisture saturation and drip discharge data at fourteen sites within the same cave studied here, between October 2011 and January 2013. A statistical approach was applied to classify the drip types and five flow regimes were identified and represented using a combined conceptual flow and box hydrological model.

The emphasis of this study is to understand the relationship between modern climatic and environmental controls on the cave

dripwaters in a region strongly influenced by ENSO; to aid in the interpretation of speleothem-based paleoenvironmental records and ultimately to develop climate proxy records from suitable speleothems (Tooth and Fairchild., 2003). Here we present the first comprehensive climate and dripwater monitoring study which commenced at three drip sites in 2006, in Harrie Wood Cave, Yarrangobilly Caves, New South Wales, Australia. This record encapsulated the last three years of the 'Millennium Drought' (1997 to 2009) which had a large impact across south-eastern Australia (CSIRO and BoM, 2015) and

the 2010/2011 La Niña event which produced widespread flooding across south-eastern Australia (CSIRO and BoM, 2015). Within this framework; spanning seven-years, we employed the results to ascertain the key hydrological processes that control the drip hydrochemistry during La Niña and El Niño events and categorised the flow regime. In doing so, we identified proxies which respond to the present day ENSO variability. Our findings form the basis for paleoclimate interpretation of speleothem trace element and $\delta^{18}$O records at Yarrangobilly Caves and are pertinent for speleothem

paleoclimate research in other ENSO dominated regions globally.

## 2      Study site and climate

Harrie Wood Cave (35° 44'S, 148° 30'E) is located in a limestone belt approximately 14 km long and 1.5 km wide along the Yarrangobilly River in the north of Kosciuszko National Park, New South Wales, Australia (Fig. 1A). The Yarrangobilly Limestone formed in the upper Silurian period from a coral reef (Worboys, 1982). The Yarrangobilly Caves system includes

over 250 independent limestone caves which began to develop in the Pleistocene (Worboys, 1982). Harrie Wood Cave is hosted within a highly fractured hard massive limestone and the drip sites are in close proximity to a fracture contact zone. The host limestone bedrock contains red/brown paleokarst features and little to no dolomite.

The cave entrance is approximately 965 m above sea level (asl) on a north-dipping steep rocky gorge. Harrie Wood is a restricted access, medium-sized south-dipping cave. The cave chamber is 80 ± 2 m in length and 34 ± 1 m deep (Nicholl,

1974). The three dripwater monitoring sites (HW1, HW2 and HW3) in this study are measured from active stalactites (Table 1) located centrally within the cave at a depth of 38 m to the surface (Fig. 1B). HW1 and HW2 were feeding actively



forming stalagmites approximately 0.5 m apart on either side of the main path. These stalagmites were removed for paleoclimate records in 2006. HW3 commenced dripping after a small twinned stalactite/stalagmite pair (column) was removed adjacent to HW2 (10-15 cm away).

Unsaturated zone hydrology of Harrie Wood Caves has recently been characterised (Markowska et al., 2015) and five dripwater regimes were identified. All flow types are fed by a theoretical soil storage and epikarst storage reservoir by fracture/fissure drainage. The five discharge flow types are as follows. Type 1 is designated mixed flow/storage connectivity (Low flow/High flow). Water at these discharge points is drained from a bulk homogenised epikarst storage reservoir. At the high flow sub-type, during periods of water excess, the epikarst store is bypassed and water is routed directly from the soil storage reservoir. At the low flow sub-type, discharge is from a pocket reservoir with a variable head within the epikarst storage reservoir. With Type 2 extreme events activated drip sites, a large intense infiltration event is required to initiate flow from the epikarst store. Type 3 are overflow sites, with discharge at the drip point fed by overflow from the pocket reservoir. Type 4 are non-linear flow sites: based on intra-karst dynamics, flow is intermittent between both storage reservoirs. Type 5 are underflow sites: during high infiltration, discharge is preferential underflow where both reservoirs are bypassed, and during baseflow conditions flow is sourced from the epikarst store. The sampling points in this study; HW1, HW2 and HW3, are situated within the transect monitored by Markowska et al., 2015 (Fig. 1B).

Surrounding vegetation consists of open snowgum (*Eucalyptus pauciflora* subsp. *Pauciflora*) and black sallee (*E. stelullata*) woodland with a snow grass (*Poa sieberi*) dominated understorey (Aplin et al., 2013). Above the cave, vegetation coverage is sparsely developed on shallow rocky soil that lack clearly defined horizons, and are dominated by angular clasts (typically 2-10 cm size) indicating mechanical weathering processes. The surface over the cave was burnt by an intense wildfire in 2003 and the shrubby vegetation that is present over the cave shows evidence of regeneration post-fire (Coleborn, 2016). There is no evidence of a distinct zone of infiltration on the surface directly above and upslope of Harrie Wood Cave. There is also no evidence of surface runoff following rain events. We interpret from our field observations that infiltration is via pervasive cracks and fissures.

Yarrangobilly Caves is part of the Australian Alps bioregion and is dominated by a montane climate (Stern et al., 2000), being characterised by mild dry summers and cold wet winters (Fig.1C). The median annual rainfall at the Bureau of Meteorology (BoM) weather station at Yarrangobilly Caves (BoM station 72141), calculated using the climatological mean 1985-2013, is 1178 ± 29 mm with a winter maximum of 349 ± 15 mm. The two dominant synoptic weather categories delivering most of this cool-season rainfall to SE Australia are cut-off low pressure systems out of the westerlies and systems from the mid latitudes including mid latitude storms and fronts (Chubb et al., 2011; Pook et al., 2014 and Callow et al., 2014). Modelled total evapotranspiration (ET) is maximum in summer and minimum in winter, and mean annual ET is 838 ± 40 mm over the period 1961-1990 (Fig. 1C). The site has a mean annual temperature of 10.5ºC, with mean maximum



temperatures in January and mean minimum temperatures in July of 27.8ºC and -1.8ºC respectively. The growing season is limited by cold winter temperatures.

Table 1. Summary of the drip point characteristics for the monitoring sites.

| Site | Description |
| --- | --- |
| HW1 | The tip of the 1 m long massive stalactite (130 mm wide) was broken, presumably when the path was constructed in ~1911 CE. A 7 cm long soda-straw stalactite has formed from the base of the massive stalactite and drips onto a 14 cm tall and 8 cm wide stalagmite developed on a flowstone on pebble ground. The stalagmite was removed in 2006. |
| HW2 | Drip emanates from a 1 m long, 110 mm wide stalactite. The 160 mm by 70 mm stalagmite was removed in 2006. |
| HW3 | Drip point formed when stalagmite from site HW2 was removed. |



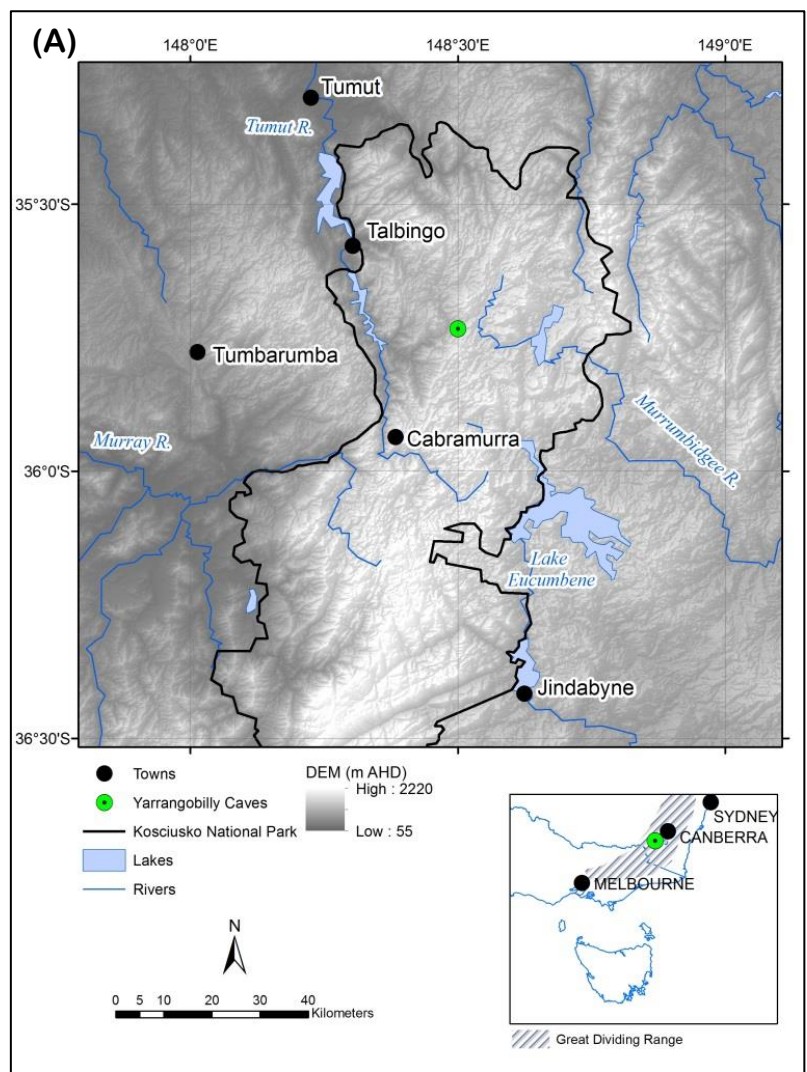

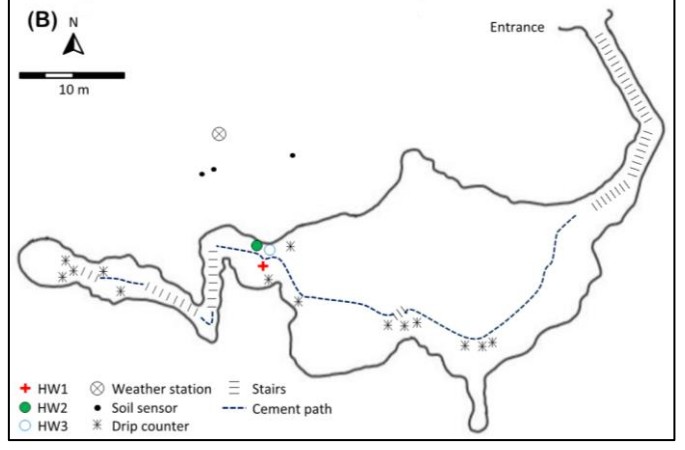

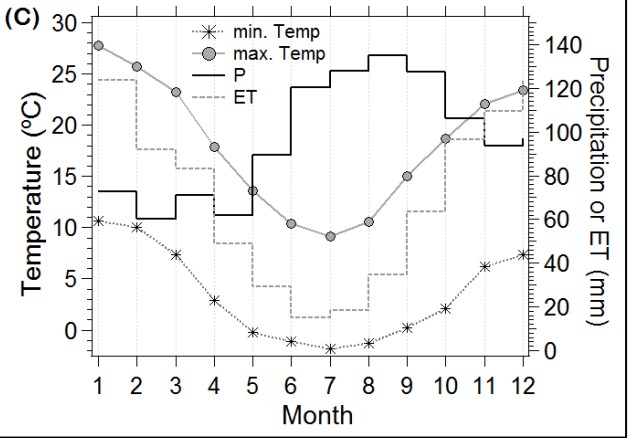





Fig. 1(A) Location of Yarrangobilly Caves, Snowy Mountains, Australia, (B) survey map of the Harrie Wood Cave system (adapted from Nicholl, 1974) showing location of the three dripwater sites (HW1, HW2 and HW3) which have been monitored since 2006; the location of automatic drip counters underneath fourteen drip sites reported by Markowska et al., 2015; and overlayed is the location of the weather station and Stevens Hydra Probe® soil sensors, and (C) site monthly mean air temperature (°C) (2006 – 2013), precipitation (mm) (1983 – 2013) and ET; the sum of transpiration and soil evaporation (1967-1990, parameter $F_{WE}$ (m day$^{-1}$) compiled by the WaterDyn model, from the Australian Water Availability Project (AWAP) database (Raupach et al. 2009, 2011)).

## 3    Methods
### 3.1    Climate record

Daily rainfall was measured from the BoM standard 203 mm rain gauge at 9 a.m. local time each morning. Provided precipitation was 2 mm or greater, an aliquot of this precipitation was collected in a 10 mL amber bottle, ensuring zero headspace, and stored for stable isotope analysis ($\delta^{18}$O, $\delta^{2}$H). Maximum and minimum air temperatures are also recorded at 9 a.m. local time using BoM standard procedures.

A network of two automatic weather stations were installed above Harrie Wood and Jillabenan Caves on the 14/10/ 2011 and 06/09/2012 respectively. Atmospheric measurements of pressure, humidity, rainfall, temperature and wind are recorded by a Davis Vantage Pro2™. Within the soil zone above each cave system, soil sensor probes were installed and buried in augered holes to a depth of 25 to 30 cm at three localities (Fig. 1B). The Stevens Hydra Probe® soil sensor measurements include temperature, soil moisture and electrical conductivity and complex dielectric permittivity (both corrected from 0 to 35°C). All data are recorded every 15 min by a Datataker DT80 data logger. A detailed description of the parameters measured by the various instrumentation and data available from the network are outlined in Supplementary EA. 1.

### 3.2    Geochemistry

Dripwaters analysed for this study were collected approximately fortnightly over the period July 2006 to December 2013 (n=468).The drip rate was measured manually as the time interval elapsed between two drips recorded using a stopwatch and the drips emanating from the stalactites accumulated in 1 L HDPE containers. Since March 2011 and sample volume permitting, in-situ field measurements of electrical conductivity (referred to 25°C, ± 1%), temperature (± 0.1°C), and pH measurements (± 0.01 pH) were made using a TPH1-MyronL TechPro II handheld meter which was calibrated using buffer solutions that were kept at cave temperature.

At each drip site, aliquots from the bulk water sample were collected and filtered through a mixed cellulose ester 0.45 μm filter and split into (i) two clean 50 ml polypropylene bottles for cation and anion analysis and (ii) a 10 ml amber glass bottle



for $\delta^{18}O$ and $\delta^2H$ analysis. On exiting the cave, samples were refrigerated, transported to the laboratory in an insulated container and maintained at 4°C until analysis.

Cation and anion analysis was conducted at the Environmental Research Chemistry Laboratory at the Australian Nuclear Science and Technology Organisation (ANSTO). Cation analysis was carried out on a Varian™ Vista Pro AX ICP-AES, prior to analysis aliquots were acidified with 1% $HNO_3$. Anion analysis was conducted on an un-acidified aliquot using a Dionex 600 Instrument with an auto suppressor. Analytical error on cation and anion analyses was ≤ 5%.

Representative limestone samples (n=7) were collected from bedrock exposures above the surface and within Harrie Wood Caves. Freshly cleaved samples were dried at 40°C and powdered using a ROCKLABS® TC-40 Tungsten Carbide ball mill. Then, 0.2 g of sample was microwave digested at 180°C for 15 minutes using aqua regia ($HCl/HNO_3$=1:3) and analysed by ICP-AES at the Environmental Research Chemistry Laboratory at ANSTO.

The isotopic composition ($\delta^{18}O$ and $\delta^2H$) of the rainfall and dripwater samples were performed on a LGR-24 d off-axis, integrated cavity output, cavity ringdown mass spectrometer at UNSW Australia. $\delta^{18}O$ and $\delta^2H$ values are reported relative to V-SMOW2. Analytical precision for $\delta^{18}O$ is 0.17‰ and 0.6‰ for $\delta^2H$.

### 3.3 Mixing and Prior Calcite Precipitation (PCP) calculations

To determine whether the geochemical evolution of the dripwaters was principally due to PCP, the hydrochemistry was assessed based on the accepted mathematical methods after Sinclair et al. (2012) and Tremaine and Froelich (2013). Firstly, to compare between the two methods, ln(Sr/Ca) (mmol mol$^{-1}$) vs. ln(Mg/Ca) (mmol mol$^{-1}$) ratios in dissolved host bedrock endmembers overlayed with the dripwater ratios was graphed. The Mg/Ca and Sr/Ca ratios during the step change in 2007, wet period and high dry-season Ca values were differentiated from the complete dataset to isolate processes affecting the dripwater chemical evolution during these periods. Then, at each drip site the slopes of the linear regression of the ln(Sr/Ca) vs. ln(Mg/Ca) (weight ratio) graph was calculated based on the model suggested by Sinclair et al. (2012); where a slope of 0.709 to 1.003, is an indicator of water/rock interactions i.e. PCP and incongruent dissolution driven processes.

## 4 Results
### 4.1 Bedrock composition

The Ca and Mg/Ca and Sr/Ca ratios for the suite of bedrock samples are listed in Table 2. Bedrock Mg/Ca ratios range from 2.6 to 15.3 mmol mol$^{-1}$ and Sr/Ca ratios lie between 0.03 to 0.32 mmol mol$^{-1}$. These low ratios indicate the host limestone rock above and within Harrie Wood Caves are a low-Mg calcite type, indicating diagenesis of the original bedrock material. We observe three groupings of samples; Ca does not vary significantly between the samples but there is an observed difference in the Mg/Ca and Sr/Ca ratios between groups. The ratios for samples (R7 – R10) are similar and there is no



difference between colour variations, the limestone conglomerate (R12) has the highest Mg content and the paleokarst samples (R13, R14) have the lowest Mg and Sr concentrations.

Table 2. Sample name, sampling location and description, Ca (mol kg$^{-1}$), Mg/Ca (mmol mol$^{-1}$) and Sr/Ca (mmol mol$^{-1}$) ratios in bedrock samples.  On the slope above Harrie Wood Caves, representative samples were collected from the soil surface. The outcrop above the cave is located 33 m west of the cave entrance along the path and at the base of the slope. Within the cave, only exposed limestone surfaces were sampled. Colours of samples are based on a visual comparison with the Munsell® rock-colour chart.

| Sample name | Sampling location; description, colour | Ca (mol kg$^{-1}$) | Mg/Ca (mmol mol$^{-1}$) | Sr/Ca (mmol mol$^{-1}$) |
|---|---|---|---|---|
| YGB_R7 | Slope above Harrie Wood Cave; 21 cm weathered limestone boulder, very light grey, contains moderate orange pink patches | 10.0 | 11.7 | 0.27 |
| YGB_R8 | Outcrop above Harrie Wood Cave; limestone, greyish black | 10.2 | 11.5 | 0.32 |
| YGB_R9 | Harrie Wood Cave, lower chamber; limestone, moderate reddish orange | 9.7 | 9.9 | 0.23 |
| YGB_R10 | Harrie Wood Cave, lower chamber;  limestone, medium dark grey | 9.7 | 8.8 | 0.24 |
| YGB_R12 | Slope above Harrie Wood Cave; limestone, conglomeratic: round very light grey fragments over 2 mm, cemented by moderate yellow finer material | 7.5 | 15.3 | 0.18 |
| YGB_R13 | Inception horizon along bedding plane in outcrop above Harrie Wood Cave; paleokarst, moderate reddish brown | 10.3 | 2.6 | 0.08 |
| YGB_R13 (duplicate) | | 10.2 | 2.6 | 0.08 |
| YGB_R14 | | 8.7 | 3.8 | 0.03 |

## 4.2    Precipitation and infiltration

The observed daily rainfall and monthly Cumulative Water Balance (CWB) at the study site and the Southern Oscillation Index (SOI) are shown in Fig. 3. The CWB represents the cumulative sum of the monthly P-AET anomalies following the method of Hurst, 1951, using the climatological mean from 1961 to 1990. The earlier half of the record overlaps the latter four years of the 'Millennium Drought', SE Australia's most persistent rainfall deficit that occurred between 1997-2009 inclusive (CSIRO and BoM, 2015). Hence the site experienced below-average rainfall from 2006 to 2009, and for much of the preceding decade. For example, from May 06 to December 06, annual rainfall totals were 52.6% below the mean attributable to a weak El Niño event (BoM, 2015a). The years 2007 to 2009 were a period of generally dry conditions, where the annual total precipitation was on average 10.3% below the 30-year average. There were exceptions of above average




monthly rainfall for February and May 2007, November 2007 to April 2008, July 2008 and in April, July and September of 2009 (Fig. 2; BoM, 2015b). However, the reduced annual totals until 2010 resulted in a decline in the calculated cumulative water balance (Fig. 3).

By contrast, in 2010 to 2012, average annual rainfall at Yarrangobilly was 152%, 135% and 115% above the mean for these

three years respectively, due to the strong 2010-12 La Niña event (BoM, 2015a). The wettest interval in our study occurred from July 2010 to March 2011 during which the calculated cumulative water balance increased (Fig. 3). The subsequent six months from May to October 2011 were relatively dry (Fig. 2), followed by a six month period of above-average monthly rainfall from October 2011 to March 2012.

Based on measurements of soil moisture saturation between 14/10/2011 and 9/01/2013, Markowska et al., 2015 interpreted

that a daily rainfall threshold range between 13 to 31.4 mm for Harrie Wood Cave was required to initiate a discharge response at fourteen monitored drip sites. During the drier 2006 to mid-2010 period of our study, there were 75 such events in total representing 30% of total raindays. By contrast, during the wetter mid-2010 to 2014 interval, there were 124 events, representing 37% of raindays.

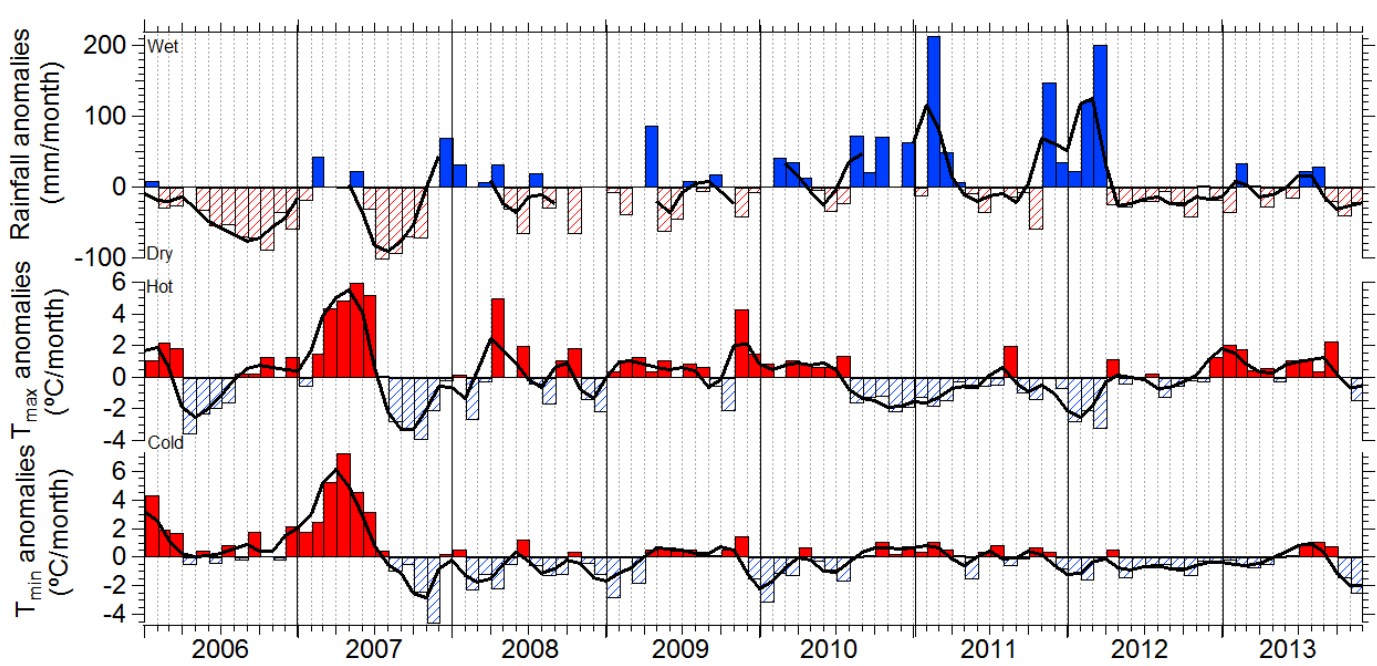

Fig. 2. Monthly rainfall, minimum temperature ($T_{min}$) and maximum temperature ($T_{max}$) anomalies at Yarrangobilly Caves (BoM station 72141) during the cave dripwater monitoring study period. The solid line represents smoothed data.





Fig. 3. The combined time series of SOI as a measure of ENSO, daily rainfall (BoM), infiltration corresponding to when rainfall exceeds 13 mm and Cumulative Water Balance (CWB), daily $\delta^{18}O_{rain}$ overlayed with the monthly Precipitation Weighted Mean (PWM), drip rate, $\delta^{18}O_{dripwater}$, Ca, Mg/Ca, Sr/Ca and Cl at drip site HW1, HW2 and HW3. Gaps within a time series indicate no available data. The dotted vertical line at July 2010 signifies the onset of the wet period. Blue and red arrows indicate an increase and decrease respectively. The pink vertical bar highlights the step change due to a flow path change.



### 4.3 Variability of $\delta^{18}$O in precipitation and dripwater

Daily rainfall, $\delta^{18}$O, monthly Precipitation Weighted Mean (PWM), as well as dripwater $\delta^{18}$O, are shown in Figure 3. The monthly PWM of $\delta^{18}$O values ranges from -13.6‰ to -2.1‰ over the sampling period, with an overall $\delta^{18}$O PWM of -6.9‰ of all rainfall events. At Harrie Wood Cave rainfall events greater than 13 mm initiate recharge (Markowska et al., 2015), the

$\delta^{18}$O PWM value from these recharge only events (-6.7‰) is not significantly different than that of all rainfall events. The monthly PWM $\delta^{18}$O values show a large 5 – 10‰ intra-annual variation following the general seasonal trend of depleted isotopic values in winter and isotopically enriched values in summer, although the winter isotopic depletion is much less pronounced in 2008, 2009 and 2011. During the dry period prior to mid-2010 the PWM was enriched (-6.2‰) compared to the 2010/12 wet period (-6.9‰).

With regards to dripwater, there is approximately 0.5 - 1‰ variability over the observation period, which is largely dampened compared to rainfall. Site HW2 had a wider range of $\delta^{18}$O values (-5.4 to -7.7‰) than site HW1 (-6.0 to -7.4‰) and HW3 (-5.7 to -7.9‰). The $\delta^{18}$O arithmetic mean values were -6.8 ± 0.3‰ for HW1 and HW2 and -6.9 ± 0.3‰ for HW3; close to the PWM (-6.9‰), suggesting significant mixing of infiltration events and that evaporation of infiltrating water is not significant. There is no clear seasonality in dripwater $\delta^{18}$O, in contrast to the rainfall $\delta^{18}$O, and there appears to be no

simple isotopic response to infiltration of winter rainfall (Fig. 3). Furthermore at all sites the mean dripwater $\delta^{18}$O values are similar prior to mid-2010 during the dry period compared to the wettest interval between mid-2010 to March 2011 (HW1: -6.7‰ cf. -7.0‰; HW2: -6.6‰ cf. -6.9‰; and HW3: -6.9‰ cf. -6.9‰), although there is a subtle trend to lighter $\delta^{18}$O values.

Figure 4 shows $\delta^{2}$H versus $\delta^{18}$O calculated Local Meteoric Water Line (LMWL; $\delta^{2}$H= (8.11±0.08) × $\delta^{18}$O + (15.9±0.5), n=415) compared with the Global Meteoric Water Line (GMWL; Rozanski et al., 1993). The LMWL intercept is greater

than the GMWL, but is in agreement to the LMWL established by Hughes and Crawford (2013) from a 3 year precipitation dataset from Big Hill in the Southern Highlands ($\delta$D = (8.10±0.12) × $\delta^{18}$O + (16.3±0.8); 188 km NE of our site, 652 m asl). Crawford et al. (2013) attributed the high intercept value to a larger contribution of moisture recycling from the land surface to the local moisture budget.





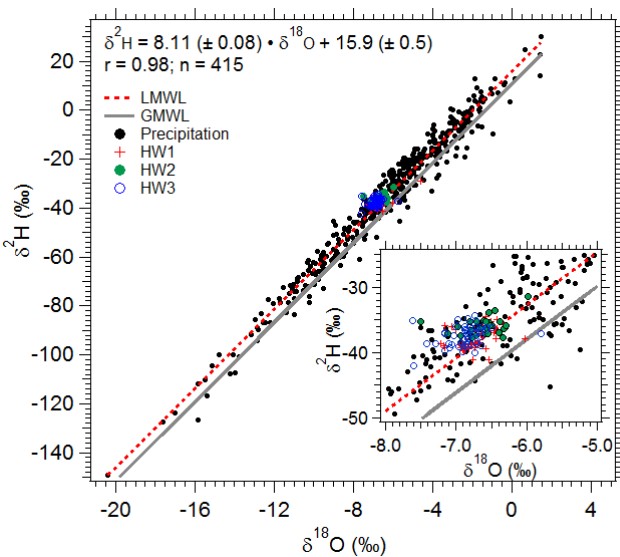

Fig. 4. Linear fit representing the LMWL for the relationship between $\delta^2H$ vs. $\delta^{18}O$ in daily precipitation samples (dashed red line) and cave dripwaters (HW1, HW2, HW3). The solid grey line denotes the position of the GMWL.

## 4.4    Drip rates

Drip rates throughout the seven year period demonstrate irregular multi-annual variations (Fig. 3). All sites remained hydrologically active, with a mean drip rate of 2.2 ± 0.6, 2.2 ± 1.7 and 3.5 ± 1.9 x $10^{-3}$ ml s$^{-1}$ at site HW1, HW2 and HW3 respectively (Table 3). Prior to mid-2010, during the long phase of below average monthly rainfall and water deficit, a base flow drip rate of 2.1 ± 0.5 x$10^{-3}$ ml s$^{-1}$ for HW1, 1.9 ± 1.4 x$10^{-3}$ ml s$^{-1}$ for HW2 and 3.1 ± 1.3 x $10^{-3}$ ml s$^{-1}$ for HW3  was maintained, only increasing slightly in response to above average monthly rainfall anomalies during this period (section 4.2);
therefore suggesting the drip sites receive flow from a storage reservoir. The increased rise in discharge is often only captured by one fortnightly sampling event, but is often present at all three sites. During the wettest interval between mid-2010 to March 2011, the mean drip rate increased to between 2.8 and 4.7 x$10^{-3}$ ml s$^{-1}$ (Fig. 3). Site HW2 is a slower dripping site compared to HW1, and HW3 is the fastest dripping stalactite. Drip rates at all three sites are mostly in phase; recording a similar pattern.

## 4.5    Dripwater chemistry

Concentrations of major cations and anions in the dripwater samples from the three drip sites are listed in Table 3. Dripwater samples from all three sites represent a Ca–HCO$_3$ dominating water type (pH 6.8–8.3) (Drever, 1982). Accordingly at drip site HW1 to HW3, Ca is the main cation in the dripwater solutions with mean concentrations of 66.4, 56.4 and 63.0 mg/l and





range from 31.1 to 96.5 mg/l. Mean concentrations of other ions are similar across the three drip sites, further supporting that the dripwaters are sourced from the same well-mixed storage reservoir.

Time series of Ca, Mg/Ca, Sr/Ca and Cl, are shown in Fig. 3. Broadly, Ca concentrations are declining from 80 to 60 mg/l at sites HW1 and HW3 from the beginning of the study until early 2008 when they become more similar to one another, but

still greater than HW2 (approx. 50 mg/l). There is a prominent rise in Ca at all sites beginning in the summer of 2008/09 and peaking in November 2009 at approximately 80 mg/l, before falling back to approximately 60 mg/l in autumn 2010. After this time, and throughout the following wetter interval, Ca concentrations at all sites become closer in value and less variable overall with some excursions to lower Ca at individual drip sites over periods of 0.5-2 months.

The prominent Ca peak in 2009 also occurs during a dry period when, Ca is increasing rather than decreasing. In this case,

this prominent peak in Ca occurs during a persistent run of above average surface temperatures dominating the entire year (Fig. 2). Specifically, an unusually warm winter was experienced in 2009 (Fig. 2), followed by the hottest November on record (BoM; 2015b). We propose that, in this interval, higher than average dripwater Ca concentrations are driven by a temperature control on net soil $CO_2$ due to respiration. We therefore propose that the warmer than average temperatures, particularly during the winter months induced higher soil $CO_2$ respiration and thereby increasing $CaCO_3$ dissolution.

Dripwater Ca returned to mean levels in January 2010 consistent with cooler temperatures (minimum temperatures were 3.1°C lower than the long-term average) restoring soil respiration conditions to previous levels (Fig. 2).

Chloride concentrations show consistent trends through time across the three sites. Chloride concentrations are highest at the beginning of our study (approx. 2.3 mg/l) and decline steadily until mid-2010 at which time, there is a more abrupt decrease in mean Cl at all three sites. After this transition, Cl concentrations are relatively stable at approx. 1.3 mg/l. Chloride is our

most conservative of the measured ions. During the transition from the relatively drier to the relatively wetter interval in our study, the abrupt decline in Cl coinciding with an increase in discharge is consistent with dilution by recharge at the onset of the wetter interval (Fig. 3). The earlier downward trend in Cl (2007 to mid-2010) appears initially inconsistent with this but we propose that reduced recharge events to the epikarst storage reservoir during this interval, resulted in a decreasing supply of Cl from the soil zone (discussed further in 5.1).

Table 3. Summary statistics of dominant ions in fortnightly dripwater samples analysed between 2006 to 2013.*n=68 samples from 15/03/2011 to 23/12/2013.

| | HW1<br>Mean ± SD (min, max) | HW2<br>Mean ± SD (min, max) | HW3<br>Mean ± SD (min, max) |
|---|---|---|---|
| Drip rate ( x $10^{-3}$ ml s$^{-1}$) | 2.2 ± 0.6 (0.8, 4.8) | 2.2 ± 1.7 (0.2, 15) | 3.5 ± 1.9 (0.06, 13.6) |
| T (°C) | 11.1 ± 0.4 (10.4, 11.9) | 11.1 ± 0.4 (10.4, 11.8) | 11.0 ± 0.4 (10.3, 11.8) |




| | | | |
|---|---|---|---|
| EC ($\mu S\ cm^{-1}$) | 311 ± 16 (275, 356) | 272 ± 33 (161, 330) | 293 ± 30 (199, 333) |
| pH | 7.82 ± 0.53 (7.17, 11.20) | 7.86 ± 0.45 (7.37, 11.00) | 7.79 ± 0.43 (7.32, 11.00) |
| **Cation** | | | |
| Ca (mg/l) | 66.4 ± 5.5 (55.7, 84.1) | 56.5 ± 7.8 (31.1, 81.4) | 63.0 ± 7.2 (41.5, 96.5) |
| Si (mg/l) | 2.3 ± 0.6 (1.4, 3.5) | 2.3 ± 0.6 (1.4, 3.8) | 2.3 ± 0.6 (1.3, 3.5) |
| *Na (mg/l) | 0.7 ± 0.04 (0.6, 0.8) | 0.7 ± 0.1 (0.6, 0.9) | 0.7 ± 0.2 (0.6, 0.8) |
| Mg (mg/l) | 0.58 ± 0.05 (0.44,0.68) | 0.58 ± 0.05 (0.45, 0.73) | 0.58 ± 0.05 (0.45, 0.82) |
| K (mg/l) | 0.12 ± 0.04 (0.07, 0.32) | 0.13 ± 0.05 (0.08, 0.35) | 0.11 ± 0.04 (0.06, 0.32) |
| Sr ($\mu g$/l) | 51.8 ± 6.7 (31.6, 61.0) | 51.0 ± 6.3 (32.3, 64.0) | 51.3 ± 6.4 (31.9, 76.0) |
| **Anion** | | | |
| Cl (mg/l) | 1.5 ± 0.4 (1.0, 2.7) | 1.6 ± 0.4 (1.1, 2.5) | 1.6 ± 0.4 (1.0, 3.0) |
| $SO_4$ (mg/l) | 0.6 ± 0.3 (0.2, 1.3) | 0.6 ± 0.2 (0.2, 1.1) | 0.6 ± 0.2 (0.2, 1.1) |
| **Elemental ratio** | | | |
| Mg/Ca (mmol $mol^{-1}$) | 14.3 ± 1.3 (10.5, 17.1) | 17.1 ± 2.1 (13.1, 30.1) | 15.3 ± 2.0 (10.7, 23.8) |
| Sr/Ca (mmol $mol^{-1}$) | 0.36 ± 0.05 (0.21, 0.44) | 0.42 ± 0.1 (0.26, 0.71) | 0.38  0.06 (0.21, 0.57) |

## 4.6    Dripwater mixing and PCP evolution

The observations above suggest processes including mixing and dilution of waters as indicated by Cl concentrations. To diagnose water/rock interactions including PCP and mixing of waters, we used a number of approaches (section 3.3). Figure 5 shows a cross-plot of ln(Mg/Ca) vs. ln(Sr/Ca) dripwater ratios presented together with the bedrock ln(Mg/Ca) and ln(Sr/Ca) ratios (after Sinclair et al., 2012 and Tremaine and Froelich., 2013). Since all three drip sites are influenced by the same processes, as the same Mg(Sr)/Ca trends are observed at all sites (Supplementary EA. 2), we present dripwater data at site HW2  for visual clarity as the processes are enhanced due to the slower discharge at this site. Many of the bedrock values are similar but lower than the dripwater ln(Mg/Ca) and ln(Sr/Ca) ratios (Fig. 5). The dripwater data themselves appear to fall into two groups: the smaller group (red squares) correspond to the beginning of the time series in 2007 and have relatively lower Sr/Ca ratios, while the larger group are the remainder of the time series for HW2 and represent the switch to relatively higher Sr/Ca. We interpret this to indicate a flow path change, which we discuss further in 5.1. All HW2 dripwater values are consistent with evolution from the bedrock values, notably R7 to R10.

Linear regressions are plotted for each group of data in Figure 5. There is some scatter of the data around these lines suggesting that more than one process may be influencing the resulting dripwater Sr/Ca and Mg/Ca ratios. During low discharge and dry rainfall conditions the dripwater composition shifts to higher Mg/Ca and Sr/Ca ratios away from the bedrock plotting in the distal outer ends, representing progressive $CO_2$ degassing due to increased PCP. During the transition period from dry to wet conditions in mid-2010 when discharge increased (blue filled circles in Fig. 5), dripwater Mg/Ca and Sr/Ca ratios plot closer to the bedrock region, indicating reduced PCP. During the high dry-season Ca values (November 2009, square window), Mg(Sr)/Ca ratios are close to the bedrock ratios. As dripwaters reached maximum Ca values,





Mg(Sr)/Ca ratios evolved towards slightly lower values, due to dissolution of the host limestone, caused by increased soil $CO_2$ from bioproductivity. An alternative hypothesis for the rise in Ca in late 2009 could be a flow path change; however the Sr/Ca and Mg/Ca data show no evidence to support this.

The calculated dripwater ln(Sr/Ca) vs. ln(Mg/Ca) (weight ratio) correlation slopes of the two groups of data at each drip site range between $0.66 \pm 0.17$ to $1.01 \pm 0.07$; within the range of predicted slopes suggested by Sinclair et al., 2012. Although a slight slope change is observed between the two flow paths, this suggests the cave dripwaters evolve under PCP.

Unlike the late 2007 step change, we observe a rise in Mg(Sr)/Ca and a drop in Ca in 2011 and 2013 (arrows; Fig. 3). These events are short lived and are most notable at one site only, there is insufficient data to evaluate this site specific change and nonetheless are features that would not be preserved in the speleothem.

Excluding 2007, over the long-term Mg values are increasing relative to Ca and there is a drift towards higher Mg/Ca values overall but less evident in the Sr/Ca time series (Fig. 3). There is a long-term rise in Mg concentrations of 0.15 ppm from 2008 to 2014. A more complete characterisation will include an investigation of aerosols and soil in a future study.

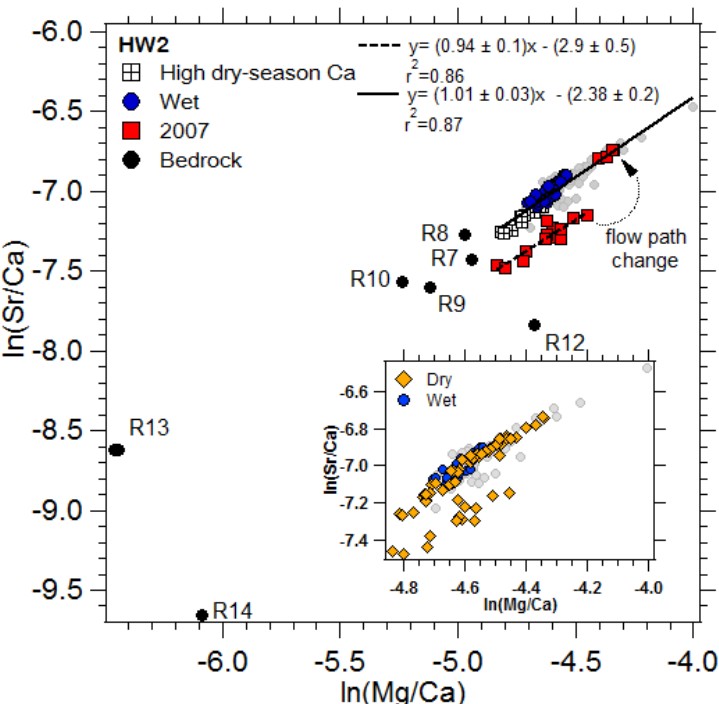

Fig. 5. Dripwater ln(Sr/Ca) vs. ln(Mg/Ca) ratios at site HW2 are graphed together with the bedrock composition (see Table 2). To enhance clarity, high dry-season Ca, wet and 2007 dripwater chemistry trends are discriminated from the complete dataset (light grey circles). The step change, offsetting between the two linear trends is indicated by a curved arrow. Inset highlights dry and wet period data.





## 5    Discussion

Speleothem geochemical proxy records archive environmental and climate signals from the surface to the cave (Fairchild and Treble, 2009). However informed paleorainfall reconstruction from speleothems requires an understanding of the drip hydrological pathway and the karst hydrogeological influences, preferably based on modern cave monitoring data from

similar climates. In this section we discuss the key results used to constrain the hydrological processses that characterise drip sites HW1 – 3 based on discharge, geochemical and stable isotope parameters in contrasting rainfall conditions, highlighting the importance of constraining the evolution of the dripwater in the context of the local environment. Multiple lines of evidence indicate the drip sites are constrained by multiple processes, which we summarise in a conceptual model. We examine the relationship between the observed hydrogeochemical changes with the climate. Our findings underlie the use of

geochemical tracers toward informing speleothem proxy records and we also discuss the implications of these results for paleoenvironmental reconstruction.

### 5.1  Hydrogeochemical processes influencing dripwater chemistry

Based on hydrogeochemical observations presented in section 4, we suggest in Harrie Wood Cave the flow paths feeding stalagmites from drip sites HW1 – HW3 are from a Type 1, mixed flow/storage connectivity low flow sub-type i.e. fracture

flow from a 'pocket reservoir' in the well-mixed epikarst storage reservoir (Markowska et al., 2015; see section 2). Our data confirm the dripwaters that precipitate speleothem calcite drain from bulk homogenised epikarst store water, the primary karst storage reservoir at Harrie Wood Caves. The results from the oxygen isotope data indicate that infiltrating water is well-mixed, because the range of dripwater $\delta^{18}O$ values is narrow in comparison to the rainfall and the arithmetic mean of the dripwaters reflects the weighted mean of precipitation (section 4.3), also from the dripwater isotope time series there is

strong buffering of the extremely low $\delta^{18}O$ winter rainfall values in 2007, 2010, 2012 and 2013 (Fig. 3). Furthermore, consistency in trace element trends among the sites (Table 3) confirms the karst waters are well homogenised.

A significant proportion of flow to the drip points is drained from a storage reservoir, owing to sustained base flow discharge levels during the weak El Niño in 2006 and the low rainfall period between 2007 and 2009 (section 4.4). Furthermore as base flow is maintained and a large infiltration event or threshold level is not required to activate the drip site, this suggests

the karst plumbing system feeding the stalagmites are not as described by Markowska et al., 2015, a Type 2; extreme events activated or Type 3 overflow site (see section 2).

Dilution of the epikarst storage reservoir was observed during the transition to the La Niña phase in 2010/11, consistent with the observation of a clear increase in discharge followed by a decrease in Cl concentrations. This further suggests that after recharge the epikarst storage reservoir volume reached close to maximum capacity maintaining "steady state"; supported by

the dripwater Mg/Ca and Sr/Ca ratios reaching mean levels indicating minimum PCP. Additionally, this indicates that



discharge at the drip sites are not from a Type 1 high flow sub-type, Type 4 or Type 5 site as during periods of water excess discharge at these sites receive flow directly from the soil storage reservoir and therefore a rise in Cl would be anticipated due to higher ET to the system as a whole, however the converse was observed.

During the dry period from 2007-mid 2010, we suggest the decreasing trend from elevated Cl levels indicates limited recharge. During the dry period we observe a decline in the CWB and reduced drip rate, consistent with fewer recharge events from the soil/vadose zone to the epikarst store and therefore decreasing dripwater Cl concentrations. Alternatively, it may also reflect higher ET in the soil/vadose zone, but this would also result in an enriched dripwater $\delta^{18}O$ signal. ET is relevant at this karst site, however we do not favour this explanation, as the similarity between dripwater $\delta^{18}O$ values during the dry period compared to the wetter period suggests evaporation of the infiltrating dripwater was not significant (section 4.3).

A flow path change at October 2007 is inferred from the shift to higher Sr/Ca ratios (Section 4.6; Fig. 5). This process occurred during the drying conditions when the CWB was decreasing (Fig. 3), but when base flow levels were maintained at all drip sites during this period. We infer that a flow path re-direction through a higher Sr/Ca endmember occurred due to soil/vadose zone drying to sustain discharge. This indicates a non-linear response of the system to progressive drying. A possible explanation is calcification of the flow path produced a threshold change causing re-routing through a higher Sr/Ca source.

Thus we deduce that the studied dripwaters are from a Type 1 mixed flow/storage connectivity low flow sub-type, the system is open to $CO_2$ and a ventilated air pocket with variable height and lower $pCO_2$ provides the potential for degassing and calcite precipitation from the dripwater (Tooth and Fairchild, 2003). Although we observed a hydrological flow path change, a distinction in dripwater Mg/Ca and Sr/Ca composition between drier and wetter periods is evident. During dry periods, the PCP mechanism was enhanced and the highest Mg/Ca and Sr/Ca ratios in this study are recorded during the El-Niño and intervals of below average rainfall, as PCP is promoted due to a dewatering of spaces. In contrast, for the duration of the strong La Niña phase in 2010/11 and above average rainfall, reduced PCP (mean Mg/Ca and Sr/Ca ratios) is noted (blue circles, Fig. 5) since a reduced reservoir head space limits degassing.

We also examine the role of calcite dissolution and soil zone $CO_2$ in more detail. Dripwater Ca concentrations reached maximum values (80 mg/l) in November 2009 (see Fig. 3); as unseasonal temperatures increased. We suggest that soil microbial production increased the $CO_2$ concentration of infiltrating waters driving $CaCO_3$ mineral dissolution (square window, Fig. 5). Presumably vadose zone $CO_2$ production also drives calcite dissolution during the dry period (Atkinson et al., 1977). During the dry period there is a long term decline in the CWB and progressive draining of the epikarst store, therefore more unsaturated zone is available for $CO_2$ production which may also lead to increased calcite solubility.





An alternative hypothesis is the Ca trend may be a fire driven process as the site was affected by fire four years before the monitoring period, which may have decreased soil $CO_2$ production, associated calcite dissolution and Ca concentrations (Coleborn et al., 2006). That effect is most likely in the first decade after the fire. However, a decreasing trend in Ca in the early monitoring period (the opposite of what might be expected) in combination with thin soil and sparse vegetation

indicates that any fire-induced soil biogeochemical changes had relatively little impact on the dripwater signature. Also, in the short term, the concentration of elements in the soil can increase in response to a fire. Considering this, during the drying trend when there is reduced recharge and based on the observed trend of the conservative tracer Cl, the effect would be an analogous decrease in Mg and Sr concentrations in the dripwaters, whereas a long-term rise in Mg is noted (section 4.6), as such we do not consider the 2003 fire event had an impact on the dripwater dataset.

Based on a 7-year observational study we have unveiled a complex non-linear geochemical response, as depicted in Fig. 6. We now attempt to discern whether these long term trends in the geochemistry are climate related. During the drying trend (2007 to mid-2010), we observed; low discharge, a flow path change, decreased Cl, enhanced and increased Ca caused by increased $CaCO_3$ solubility due to higher soil $CO_2$ and bioproductivity. This is a response consistent with reducing recharge conditions. During the wetter period we also observed trends consistent with increased recharge; increased discharge,

dilution and reduced PCP. Therefore, we interpret these as evidences of climatic induced changes. Over the long-term, drip discharge and hydrogeochemical variations between the relatively dry and wet period are driven by variations in water availability due to ENSO and are of paleoclimatic relevance.

We have confidently constrained the possible hydrological processes which can occur in a karst system situated in a region greatly impacted by ENSO episodes. We attribute this to long-term monitoring that encapsulated the shift between the two

extremes of ENSO, namely from dry El Niño to wet La Niña conditions. Based on our findings and with the knowledge that the impact of each ENSO episode on Australia varies, we have identified how Yarrangobilly Caves responds to a changing climate signal (rainfall), and through geochemical proxies ($\delta^{18}O$, Ca, Cl and Mg/Ca and Sr/Ca ratios) identified how this response has been transformed and is recorded in cave dripwaters over time. These results have important implications in informing paleoclimate findings from speleothem archives which we discuss further in section 5.2.





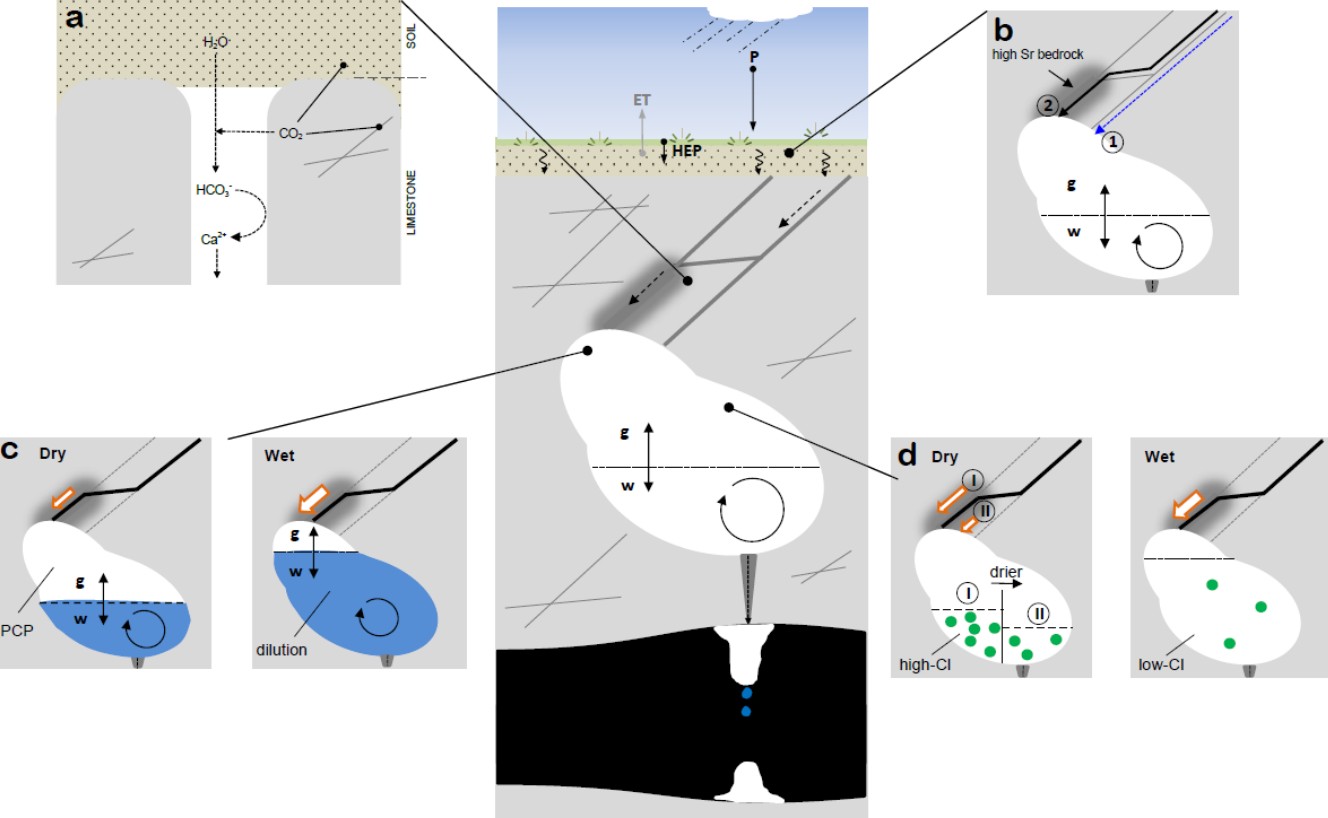

Figure 6. Conceptual model summarising key process affecting dripwater hydrochemical variations at Harrie Wood Cave. Shown are Precipitation (P), Evapotranspiration (ET) and Hydrologically Effective Precipitation (HEP), infiltrations corresponding to when rainfall exceeds 13 mm. Stalactites from drip sites HW1–HW3 are fed by fracture flow from a ventilated well-mixed pocket reservoir with a variable head within the epikarst. (a) Soil and unsaturated/vadose zone $CO_2$ drive $CaCO_3$ dissolution, increasing dripwater Ca. (b) The host bedrock varies geochemically and the dark grey shading along the fracture represents bedrock of higher Sr concentrations. A flow path change from 1 to 2 (October 2007) results in dripwaters being enriched in Sr. The size of flow arrows in (c) and (d) qualitatively correspond to recharge to the epikarst storage reservoir. (c) PCP within the store is enhanced during dry periods as the water level (w) is low, fairly constant and in contact with a greater ventilated gas phase (g). Dilution occurs during wet intervals due to greater inflow and PCP decreases as a reduced reservoir head space limits degassing (w>g). (d) During dry periods, the response to reduced inflow (I to II) is a decrease in Cl concentrations within the storage reservoir and therefore dripwater. In wet periods, Cl is flushed from the soil/vadose zone and then diluted within the store, resulting in a sharp decline in dripwater Cl.



## 5.2 Implications for speloethems as paleorainfall recorders

Our results provide a foundation to inform speleothem paleoclimate records where dripwater compositions are influenced by multiple processes. During our studied interval, element concentrations and ratios in the dripwater were driven by climate (ENSO) and karst hydrological processes. Based on drip characterisation of sites HW1 – 3, Mg/Ca variations in the conjugate stalagmite could in principle, be successfully applied to construct paleo-rainfall conditions (Tooth et al., 2003; Markowska et al., 2015). We have demonstrated that dripwater Mg/Ca and Sr/Ca ratios are a relative measure of modern rainfall variability. Therefore in a highly resolved speleothem time series, we anticipate displacements of $[Mg/Ca]_{calcite}$ and $[Sr/Ca]_{calcite}$ from a baseline ratio (bedrock) to higher and mean ratios, will differentiate between dry and wet periods, respectively. McDonald et al., 2004 also identified a relationship between dripwater Mg/Ca and Sr/Ca ratios and drought in Wombeyan Caves, NSW; however their interpretation of the trace element stalagmite record varies slightly. At Wombeyan, PCP is a more dominant control on the karst hydrochemistry, although it is in the same climate region, but a warmer site with greater evapotranspiration and a longer growing season. Consequently, we would expect that the resultant dripwater (and stalagmite calcite) would be enriched in Mg/Ca and Sr/Ca. As such we expect to observe a greater displacement of Mg/Ca ratios away from the bedrock composition during any given dry phase at Wombeyan Caves, in comparison to a corresponding record at Yarrangobilly Caves. For wet periods, no La Niña events occurred during their study period (July 2001 – January 2004); therefore McDonald et al., (2004) could only speculate Mg/Ca calcite ratios would shift towards bedrock values due to a decreased effect of PCP. In contrast, extended monitoring through the La Niña mode of an ENSO cycle provides us with a more informed interpretation; whereby we anticipate La Niña events in the stalagmite trace element time series may be resolved from baseline ratios by a shift to mean ratios.

## 6 Conclusions

This research targeted the Snowy Mountains region of the Australian Alps; a key water resource region in SE Australia where rainfall variability and therefore future water resource availability in this climatically sensitive region is uncertain. Our results have advanced the scientific knowledge of the in-cave drip response to modern-ENSO variability, via rainfall, and therefore enabled identification of the most reliable geochemical proxies (trace element concentrations and $\delta^{18}O$) for paleoclimate reconstruction from speleothem archives within Harrie Wood Caves and in the region.

This study underpins the importance of extended monitoring through a modern ENSO cycle and we document a site where a number of concurrent hydrological processes are occurring occasioning a complex non-linear geochemical dripwater signature. The local karst hydrogeological processes which influence trace element and $\delta^{18}O$ proxies in dripwater, and therefore signatures in the speleothem calcite, were constrained. We identified discharge at drip sites HW1 –HW3 is from a well-mixed pocket reservoir in the Epikarst Storage Reservoir.





Interpretation of dripwater Ca, Cl, Mg/Ca and Sr/Ca ratios during contrasting rainfall conditions allowed the following processes to be constrained. During the El Niño and dry periods, enhanced PCP resulted in higher Mg/Ca and Sr/Ca ratios. While during the La Niña and wet phase, Cl concentrations and discharge were used to constrain the process of dilution and reduced PCP controlled drip chemistry. However we found a number of non-linear responses embedded in a linear drying climate trend. An interpretation of dripwater Sr/Ca vs. Mg/Ca ratios compared to different end-member bedrock ratios showed a shift to higher Sr values, suggesting a flow-path change. Decreasing dripwater Cl levels indicated reduced recharge of the epikarst storage reservoir, and an unexpected rise in the Ca time series was shown to be caused by carbonate dissolution due to an increase in soil and epikarst/vadose zone $CO_2$ concentrations. The data presented here highlights the complex interplay of a dripwater signal in response to the climate signal, this has been achieved only through long term monitoring over a seven-year (2007-2013) period. A conceptual model was constructed illustrating these key processes controlling the dripwater composition.

These processes are shown to be linked to climate induced changes and propose Mg/Ca and Sr/Ca variations in the conjugate stalagmite could in principle, be successfully applied to construct paleorainfall conditions. This study has extended our understanding of changing climatic controls on proxy variations within the Harrie Wood Cave system and therefore provides a benchmark for its application regionally and globally, when using speleothems for paleoclimate reconstruction.

**Data availability**

Time series of rainfall and dripwater data used in this study are available upon request from the corresponding author. The Southern Oscillation Index monthly data are publicly available (http://www.bom.gov.au/climate/current/soihtm1.shtml). This work used data acquired from the Australian Water Availability Project (AWAP), AWAP model results can be accessed by contacting Peter Briggs (http://www.csiro.au/awap/; Peter.Briggs@csiro.au).

**Author contribution**

C.V.T. conceptualised the research, collected data, conducted all data analysis and interpretation, generated graphs and conceptual model and wrote the manuscript. P.C.T. and A.B. provided guidance, reviewed and edited the manuscript in their function as my supervisors, as did I.F. S.H. installed and maintained the weather stations and assisted with generating Fig. 1(A). R.R. collected the fortnightly dripwater samples for this study. M.M. performed the stable isotope analyses. J.M. set up the dripwater monitoring study in 2006.



## Acknowledgements

The authors are grateful to Suzanne Hollins for supporting this research. We thank Jagoda Crawford and Darrell Tremaine for useful discussion. Henri Wong, Chris Vardanega, Robert Chisari and Barbora Gallagher are thanked for their assistance with sample analysis. George Bradford and the staff at Yarrangobilly Caves and NSW NPWS are also thanked for their

dedication and on-going field support and access permission. Silvia Frisia and Andrea Borsato are thanked for their assistance in the field regarding the geology of karst in the Snowy Mountains alpine region. Peter Briggs and Alan Griffiths are acknowledged for providing the AWAP data. PCT acknowledges the support of a Land & Water Australia grant (project number ANU52) for this study.

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
