# Peer review of "ENSO - cave dripwater hydrochemical relationship: a 7-year dataset from SE Australia"

_Hydrology and Earth System Sciences, 2016_

## Referee Comment (RC1) · Anonymous Referee #1 · 7 Jul 2016

Interactive comment on: "ENSO – cave dripwater hydrochemical relationship: a 7-year dataset from SE Australia" by C. V. Tadros et al.

Anonymous Referee # 1: This manuscript presents a 7-year dripwater monitoring study from Harrie Wood Cave in SE Australia. Several dripwater parameters were monitored for seven year in a two weeks rhythm. These results were used to interpret the dripwater interactions and processes evolving the dripwater composition in the epikarst. This is linked to the outside precipitation, which is extremely influenced by the ENSO in this region. This gives the opportunity to interpret Mg/Ca and Sr/Ca from stalagmites of this cave in the case of ENSO variability. It is a very interesting story and a nice data set, which is a good contribution to the scientific progress. Especially, it is nice to see that now more and more long-term monitoring studies are published. Further, the manuscript is well written and structured, but there are some points were the

manuscript should be improved. In the following I will list comments and suggestions, as well as technical improvements for the manuscript. The manuscript needs minor revisions before accepting it for publication.

Comments in chronological order: General comment: On the one hand you wrote Yarrangobilly Caves and on the other hand Harrie Woods Cave. If I understand it correct, the Harrie Woods Cave is one cave in an area with a lot of caves, which are called the Yarrangobilly Caves. This is sometimes a bit confusing in the paper. I think all your data are from Harrie Woods Cave. Therefore, I would recommend to always write Harrie Woods Cave in the manuscript and only write in the "Study site" section that this caves belong to a bigger cave region, which is called Yarrangobilly Caves. Please, also mark Harrie Woods Cave in Figure 1 instead of Yarrangobilly Caves. That makes it easier to understand for the reader.

P1, L24: Is it correct that soil $CO_2$ production increases during drier periods? In some cases this occurs when there is more precipitation.

P2, L20: Please delete the "dissolution" behind "differential" and set a comma behind the second "dissolution" in that line.

P2, L21: Please set a comma behind the citation parentheses.

P3, L2: Please change "Callow et al., 2014" into "Callow et al. (2014)".

P3, L6: Please change "Markowska et al., (2015)" into "Markowska et al. (2015)".

P3, L7-8: Please clarify that these five flow regimes are identified by Markowska et al. (2015), it is at the moment not clear from this sentence.

P3, L13: Please put "New South Wales, Australia" in parentheses.

P3, L17-18: Please delete "In doing so, we identified proxies which respond to the present day ENSO variability." because this fits better to the results part.

P3, L31: Please change "…located centrally within the cave at a depth of 38 m to the

surface (Fig. 1B)." to ""…located centrally within the cave (Fig. 1B) at a depth of 38 m to the surface.".

P3, L30 – P4, L3: These lines should fit better to the material part.

P4, L2: Please replace "paleoclimate records" with "paleoclimate studies".

P4, L4-15: It would be a better structure putting this paragraph and the also before mentioned sentence about the different drip sites to an one paragraph. Making 2.1 Study site and climate and 2.2 Dripsite settings.

P4, L15: Please replace "Markowska et al., 2015" with "Markowska et al. (2015)".

P4, L25: Is "median annual rainfall" correct? I think it should be the annual precipitation sum. Please check this.

P4, L31: Please delete "(Fig. 1C)" in this line, because the mean annual ET is not shown in this figure, but the monthly ET.

P5, L1: Please set a comma in front of "respectively".

P5, Table1: Why does the drip site HW3 formed when HW2 stalagmite was removed? I do not understand this please give more information. Was something broken?

P6, Figure 1B: Please find another signature for HW3. Perhaps a colour filled circle. It is not very visible and the drip sites are the important points.

P6, Figure 1C: Please replace "or" at the second y-axes with "and".

P7, L3-4: Please replace "Markowska et al., 2015" with "Markowska et al. (2015)".

P7, L9: Please replace "Climate record" with "Meteorological data".

P7, L15: Please add a comma in front of "respectively".

P7, L15: Did you measure wind speed and/or direction? Please add this information.

P8, L7: Please replace "…exposures above the surface…" with "… exposures above

the cave".

P8, L25: Please add "concentration behind "Ca", because this is no ratio.

P9, L1-2: Please change "... highest Mg content and the paleokarst samples (R13, R14) have the lowest Mg and Sr concentrations." to "... highest Mg/Ca ratio and the paleokarst samples (R13, R14) have the lowest Mg/Ca and Sr/Ca ratios." , because you gave ratios in Table 2.

P9, L3: Please add "concentration" behind "Ca", because this is no ratio.

P9-11: Please change Figure 2 and 3, because you first cited Figure 3, therefroe it should be Figure 2.

P9, L 9: Is the Cumulative Water Balance the same as the effective infiltration? Therefore, precipitation minus evapotranspiration? It is not familiar to me. Perhaps you could use another word for this like infiltration or give a bit more explanation on this.

P10-11: Please give the panels in Figure 2 and 3 some a), b), c) and so on and also cite directly the different panels. That make it easier for the reader to find the right panel in the figure.

P10, L7-8: I could only see five wetter month from November 2011 to March 2012. October 2011 is quite the driest month of 2011. Please check this and clarify it.

P10, L9: Please replace "Markowska et al., 2015" with "Markowska et al. (2015)".

P10, L16: Please give some information how the data were smoothed. (10 point, 2 point filter, smoothed from the daily data?).

P11: In panel b) the bars for the rain in grey are hard to see, please take another colour. In panel c) the dots for the d18O of the rain water are as well hard to see. Do the PWM do not have a unit? Why do you give (wt) as a unit for Mg/Ca and Sr/Ca for the water samples? It is a bit confusing to me, because it is liquid. Should be volume percent?
[Figure]

P12, L2: Please add "rain water" in front of d18O.

P12, L 5: Please change "...these recharge only events...." to "...these recharge events...".

P13, L7: Please add a comma in front of "respectively".

P13, L4-14: Could you perhaps add the drip characteristics such as seepage flow and seasonal drip to this paragraph?

P13-14, L17-1: I do not understand this sentence. Please rephrase it.

P15, L10-11: Are not also the Mg/Ca higher? This is not completely clear to me. Please give some more information.

P16, L5: Please replace "Sinclair et al., 2012" by "Sinclair et al. (2012).

P16, L12: 2014? You only show Mg data in Figure 3 until 2013.

Figure 5: The light grey circles are very hard to see. Please choose another colour for better visibility.

P17, L25: Please replace "Markowska et al., 2015" by "Markowska et al. (2015).

P1, L20: Please replace "composition" by "ratio".

P20, Figure 6: The indication of a and b in the middle picture is switched. Please correct this.

P20, L4: Do you mean infiltration with "Hydrological Effective Precipitation"? If this is the case infiltration is much handier.

P21, L9: Please replace "McDonald et al., 2004" by "McDonald et al. (2004)".

P21, L16: Please delete the comma in front of "(2004)".

P23, L2: Is Barbora Gallagher correct? Or should it be Barbara?

---

## Referee Comment (RC2) · Anonymous Referee #2 · 23 Jul 2016

The karst hydrological and geochemical processes are strongly affected by ENSO in the SE Australia. This paper links the El Niño and La Niña phases to the hydrochemical variations of cave dripwater, Harrie Wood Caves, SE Australia. Based on the El Niño and La Niña phases, the authors investigate the hydrochemical processes by using a 7-year long rainfall isotope 18O and dripwater Ca, Cl, Mg/Ca, and Sr/Ca dataset from three drip sites, in Harrie Wood Caves. Results show that during the El Niño and dry periods, enhanced Prior Calcite Precipitation (PCP) resulted in higher Mg/Ca and Sr/Ca ratios. While during the La Niña phase where dilution dominated and reduced PCP. The research is interesting and worth to publish in Hydrology and Erath System Science after minor revision.

Comments: 1. You use Yarrangobilly Caves, and Harrie Wood Caves in text. Please clarify the relation between the two caves in text, and illustrate in Fig.1.

2. There are three dripwater monitoring sites (i.e. HW1, HW2, and HW3), and the distance between the three sites are less than 10m (Fig.1B). The three drip sites belong to discharge flow Type 1, mixed flow/storage connectivity. But the observation data of Ca, Mg/Ca, and Sr/Ca for HW2 are very different from HW1 and HW3. Why?

—————————————————————

---

## Author Comment (AC1) · 4 Aug 2016

August 4, 2016

Dr Bill Hu, Editor

HESSD

Dear Dr Hu,

Please find attached the author response to the referees comments of the manuscript "ENSO - cave dripwater hydrochemical relationship: a 7-year dataset from SE Australia" by Tadros et al. for your consideration. On behalf of the authors I would like to take this opportunity to sincerely thank them for their time and effort in the careful reading of our manuscript, we acknowledge their comments and consider these will improve the quality of our manuscript.

Both referees queried the site nomenclature, but referee 1 is correct in surmising that Harrie Wood is one of the caves in the Yarrangobilly Group. Both referees also commented on wanting further information on the drip site setting. Other comments were routine corrections of presentational errors and clarification on a few points.

Herein, I explain how we will revise the paper based on those comments and recommendations. Please note, referee comments are in blue font, italicised and underlined and the author responses are in black font. We believe we have addressed all comments raised by the two reviewers of the manuscript.

A revised version will be provided if decision has been made to accept the manuscript. I look forward to hearing from you regarding our submission.

With kind regards,

For the authors,

Carol Tadros

**Anonymous Referee #1 comment**

*C2, General comment: "… I would recommend to always write Harrie Woods Cave in the manuscript and only write in the "Study site" section that this caves belong to a bigger cave region, which is called Yarrangobilly Caves…"*

We agree, and for the sake of clarity, in the revised manuscript, we will replace Yarrangobilly Cave with Harrie Wood Cave throughout the manuscript where we are referring specifically to the cave [P1, L17; P3, L13; Fig. 1(A); P7, L1; P19, L21; and P21, L15].

As suggested by the referee, the exception is the study site section and P3, L24. Therefore in the revised manuscript we propose it to read: "…Harrie Wood Cave is located within the Yarrangobilly Caves system, which includes over 250 independent limestone caves …"

When we are referring to the karst system as a whole unit [P2, L5; P3, L19; P4, L24] we will be more specific and use "Yarrangobilly Caves system" and "Yarrangobilly Caves (BoM station 72141)" when we are referring to the Bureau of Meteorology station (72141) or data obtained from the station [P4, L26; P10, L4; P10, L15]; as this is the official name of the BoM station.

*C2, General comment: "… Please, also mark Harrie Woods Cave in Figure 1 instead of Yarrangobilly Caves…"*

In the revised manuscript, Harrie Wood Cave will replace Yarrangobilly Caves in Fig. 1(A).

**MINOR TECHNICAL CORRECTIONS:**
All the below suggestions from P2, L20 to P21, L16 refer to minor text revisions and will be corrected in the revised manuscript.

*P2, L20: Please delete the "dissolution" behind "differential" and set a comma behind the second "dissolution" in that line.*

*P2, L21: Please set a comma behind the citation parentheses.*

*P3, L2: Please change "Callow et al., 2014" into "Callow et al. (2014)".*

*P3, L6: Please change "Markowska et al., (2015)" into "Markowska et al. (2015)".*

*P3, L13: Please put "New South Wales, Australia" in parentheses.*

*P3, L17-18: Please delete "In doing so, we identified proxies which respond to the present day ENSO variability." because this fits better to the results part.*

*P3, L31: Please change "…located centrally within the cave at a depth of 38 m to the surface (Fig. 1B)." to ""…located centrally within the cave (Fig. 1B) at a depth of 38 m to the surface.".*

*P4, L2: Please replace "paleoclimate records" with "paleoclimate studies".*

*P4, L15: Please replace "Markowska et al., 2015" with "Markowska et al. (2015)".*

*P4, L31: Please delete "(Fig. 1C)" in this line, because the mean annual ET is not shown in this figure, but the monthly ET.*

*P5, L1: Please set a comma in front of "respectively".*

*P6, Figure 1C: Please replace "or" at the second y-axes with "and".*

*P7, L3-4: Please replace "Markowska et al., 2015" with "Markowska et al. (2015)".*

*P7, L9: Please replace "Climate record" with "Meteorological data".*

*P7, L15: Please add a comma in front of "respectively".*

*P8, L7: Please replace "...exposures above the surface..." with "...exposures above the cave".*

*P8, L25: Please add "concentration behind "Ca", because this is no ratio.*

*P9, L1-2: Please change "... highest Mg content and the paleokarst samples (R13, R14) have the lowest Mg and Sr concentrations." to "... highest Mg/Ca ratio and the paleokarst samples (R13, R14) have the lowest Mg/Ca and Sr/Ca ratios." , because you gave ratios in Table 2.*

*P9, L3: Please add "concentration" behind "Ca", because this is no ratio.*

*P9-11: Please change Figure 2 and 3, because you first cited Figure 3, therefore it should be Figure 2.*

*P10, L9: Please replace "Markowska et al., 2015" with "Markowska et al. (2015)".*

*P12, L2: Please add "rain water" in front of d18O.*

*P12, L 5: Please change "...these recharge only events... ." to "...these recharge Events ...".*

*P13, L7: Please add a comma in front of "respectively".*

*P16, L5: Please replace "Sinclair et al., 2012" by "Sinclair et al. (2012).*

*P17, L25: Please replace "Markowska et al., 2015" by "Markowska et al. (2015).*

*P20, Figure 6: The indication of a and b in the middle picture is switched. Please correct this.*

*P21, L9: Please replace "McDonald et al., 2004" by "McDonald et al. (2004)".*

*P21, L16: Please delete the comma in front of "(2004)".*

**CLARIFICATIONS TO TEXT**
*P1, L20: Please replace "composition" by "ratio".*
Thank you for your suggestion however we believe composition is more accurate than ratios, as variations in both concentration and ratios are being referred to here.

*P1, L24: Is it correct that soil CO2 production increases during drier periods? In some cases this occurs when there is more precipitation.*
Thank you for drawing our attention to this oversight, we intended to write "… in response to warmer than average temperatures…" as documented in P14, L13. This will be corrected.

*P3, L7-8: Please clarify that these five flow regimes are identified by Markowska et al. (2015), it is at the moment not clear from this sentence.*
For clarification we propose the sentence "…A statistical approach was applied to classify the drip types and five flow regimes were identified …" on P3, L7-8 will read:

"Markowska et al. (2015) applied a statistical approach to classify the drip types and identified five flow regimes which were represented using a combined conceptual flow and box hydrological model.".

*P3, L30 – P4, L3: These lines should fit better to the material part. See response to comment below*

*P4, L4-15: It would be a better structure putting this paragraph and the also before mentioned sentence about the different drip sites to an one paragraph. Making 2.1 Study site and climate and 2.2 Dripsite settings.*
We agree and in the revised manuscript Section 2 will be re-structured under section heading 'Study Area'. Study site and climate will become sub-section 2.1 and as suggested by the referee, P3, L30 – P4, L3 and P4, L4-15 will appear under subheading 2.2 Drip site setting.

*P4, L25: Is "median annual rainfall" correct? I think it should be the annual precipitation sum. Please check this.*

Median annual rainfall is correct. Long term rainfall data observations at Yarrangobilly Caves (BoM station 72141) were used to calculate the median annual rainfall, based on data between 1985-2013. The median was used as a measure of the 'average' rainfall, since extreme rainfall events bias the arithmetic mean greater than the median.

*P5, Table1: Why does the drip site HW3 formed when HW2 stalagmite was removed? I do not understand this please give more information. Was something broken?*

In order to remove HW2, a small adjacent calcite column that had formed from a fused stalactite-stalagmite pair approximately 10-15 cm from HW2 had to be removed also. This resulted in re-invigoration of the drip point that had formed the column, which we included in our sampling program and refer to as HW3. This information will be added to the text on P4, L2. In addition, supporting material will be provided to the reader in the form of a photograph and drip rate field observations of site HW2 prior to the removal of stalagmite HW2 (constituting Supplementary EA. 1).

*P6, Figure 1B: Please find another signature for HW3. Perhaps a colour filled circle. It is not very visible and the drip sites are the important points.*

We agree and drip site HW3 will be represented in Fig. 1B as a filled blue circle.

*P7, L15: Did you measure wind speed and/or direction? Please add this information.*

Both wind speed and direction were measured (please see Supplementary EA.1.). This information will be added so that the sentence on P7, L15 will read:

"… Atmospheric measurements of pressure, humidity, … and wind speed and direction are recorded by a Davis Vantage Pro2™.".

*P9, L 9: Is the Cumulative Water Balance the same as the effective infiltration? Therefore, precipitation minus evapotranspiration? It is not familiar to me. Perhaps you could use another word for this like infiltration or give a bit more explanation on this.*

We will provide a better explanation of this and propose it will be corrected in the revised manuscript to read:
"The CWB represents a residual mass curve to show the cumulative monthly water budget trends, following the method of Hurst, 1951. It is calculated as the cumulative sum of the monthly P-ET anomalies from the climatological mean (1961-1990).".

*P10-11: Please give the panels in Figure 2 and 3 some a), b), c) and so on and also cite directly the different panels. That make it easier for the reader to find the right panel in the figure.*

We much appreciate the reviewer's suggestion and agree the visual clarity of Fig. 2 and Fig. 3 will be improved by inserting letter labels.

*P10, L7-8: I could only see five wetter month from November 2011 to March 2012. October 2011 is quite the driest month of 2011. Please check this and clarify it.*

We much appreciate the reviewer's observation. This was a typographical error and P10, L7-8 should read:

" … followed by a five month period of above-average monthly rainfall from November 2011 to March 2012 (Fig. 3E).". This will be corrected in the revised manuscript.

*P10, L16: Please give some information how the data were smoothed. (10 point, 2 point filter, smoothed from the daily data?).*

The solid line is the result of applying a binomial smoothing, 1 pass Gaussian filter to the monthly data. This information will be inserted in the revised manuscript.

*P11: In panel b) the bars for the rain in grey are hard to see, please take another colour. In panel c) the dots for the d18O of the rain water are as well hard to see. Do the PWM do not have a unit? Why do you give (wt) as a unit for Mg/Ca and Sr/Ca for the water samples? It is a bit confusing to me, because it is liquid. Should be volume percent?*

We agree. In the revised manuscript the colours for rain and rain $\delta^{18}O$ will be changed. The units for PWM are "‰" and this will be added.

The scientific units for concentration of solutes in a solvent are correctly expressed as mg l$^{-1}$ (ppm or "parts per million"). As a consistent comparison with the dripwater Ca concentrations (mg l$^{-1}$) we reported weight ratios (mg mg$^{-1}$) for Mg/Ca and Sr/Ca.

In order to further improve the clarity of the axis labels on Figure 2, we will also change the units from mg mg$^{-1}$ to mg g$^{-1}$ to avoid the use of scientific notation on the Y-axis.

*P13, L4-14: Could you perhaps add the drip characteristics such as seepage flow and seasonal drip to this paragraph?*

Reviewer 1 refers to the Smart and Friederich (1987) classification system which characterises drip sites based on discharge response to recharge events and discharge variability. McDonald and Drysdale (2007) identified that this classification system is not applicable to southeast Australia or regions where ENSO is the main regulator of recharge and Markowska et al. (2015) showed that this classification was not appropriate for Harrie Wood Cave. As such we prefer not to classify the drip sites in this study using this classification system.

**References:**

Smart PL, Friederich H. 1987. Water movement and storage in the unsaturated zone of a maturely karstified aquifer, Mendip Hills, England. In Proceedings of the Conference on Environmental Problems in Karst Terrains and their Solutions, 28–30 October 1986, Bowling Green, Kentucky. National Water Wells Association: 57–87.

McDonald J, Drysdale R. 2007. Hydrology of cave drip waters at varying bedrock depths from a karst system in southeastern Australia. Hydrological Processes 21: 1737–1748. DOI:10.1002/hyp.6356.

*P13-14, L17-1: I do not understand this sentence. Please rephrase it.*

We will rephrase the sentence on P13-14, L17-1 and propose it will be corrected in the revised manuscript to read:

"Dripwater samples from all three sites represent a Ca–HCO$_3$ dominating water type (pH 6.8–8.3) (Drever, 1982), accordingly Ca is the main cation in the dripwater solutions. The mean Ca concentrations observed at drip sites HW1 to of HW3 are 66.4, 56.4 and 63.0 mg l$^{-1}$, respectively and range from 31.1 to 96.5 mg l$^{-1}$".

*P15, L10-11: Are not also the Mg/Ca higher? This is not completely clear to me. Please give some more information.*

The observed step change indicates a change in the initial composition of the chemistry at equilibration based on the Sinclair et al. (2012) examination of these processes. We will indicate this explicitly in the text and it will be corrected in the revised manuscript to read:

"The dripwater data … ratios. If the geochemical evolution of the dripwaters was solely due to PCP, Sinclair et al. (2012) mathematically demonstrates that the dripwater ratios would produce a straight line correlation of known slope through the whole dataset (section 3.3). The observed step change indicates a change in the initial composition of the chemistry at equilibration. We interpret this to indicate a flow path change …"

*P16, L12: 2014? You only show Mg data in Figure 3 until 2013.*

This was a typographical error and will be corrected to read 2013.

*Figure 5: The light grey circles are very hard to see. Please choose another colour for better visibility.*

For enhanced visibility this will be corrected.

*P20, L4: Do you mean infiltration with "Hydrological Effective Precipitation"? If this is the case infiltration is much handier.*

We agree and in the revised manuscript "Infiltration" will replace "Hydrological Effective Precipitation".

*P23, L2: Is Barbora Gallagher correct? Or should it be Barbara?*

This has been checked and is correct; it is the way Barbora spells her name.

**Anonymous Referee #2 comment**

*Comment: 1. You use Yarrangobilly Caves, and Harrie Wood Caves in text. Please clarify the relation between the two caves in text, and illustrate in Fig.1.*

The same comment was raised by Anonymous Referee #1 and we would like to refer you to our reply above in C2, General comment.

*Comment: 2. There are three dripwater monitoring sites (i.e. HW1, HW2, and HW3), and the distance between the three sites are less than 10m (Fig.1B). The three drip sites belong to discharge flow Type 1, mixed flow/storage connectivity. But the observation data of Ca, Mg/Ca, and Sr/Ca for HW2 are very different from HW1 and HW3. Why?*

Indeed, in the results section we document the observations that drip site HW2 is a slower dripping site (P13, L12) and the Ca concentrations at site HW2 are lower compared to HW1 and HW3 (P14, L5) although the Cl concentration trends are consistent (P18, L19). However, we overlooked addressing this in the Discussion and are pleased the referee has pointed this out. Therefore in the revised manuscript we will insert the following interpretation as a paragraph in the Discussion (P18, L17):

"The three dripwater monitoring sites (HW1, HW2, and HW3) are located within a small area (Fig. 1B), however there are differences in the chemistry, notable in 2007 during the earliest interval of the dry period. In particular a lower drip rate (section 4.4) and a lower Ca concentration and higher Mg/Ca and Sr/Ca ratio (section 4.5) of site HW2 compared to the neighbouring site HW3 and HW1, which are only 10 – 15 cm away. Equivalent Cl concentrations over this period (section 4.5) suggest differences have subsequently arisen in the carbonate chemistry. A potential explanation for this is that all three drip sites are fed by the same chemistry, but the lower drip rate at site HW2 is consistent with greater in cave PCP on the stalactite tip, inducing a lower Ca concentration and enhancing PCP (Treble et al., 2015)."

**Reference:**
Treble, P. C., Fairchild, I. J., Griffiths, A., Baker, A., Meredith, K. T., Wood, A., and McGuire, E.: Impacts of cave air ventilation and in-cave prior calcite precipitation on Golgotha Cave dripwater chemistry, southwest Australia, Quaternary Sci. Rev., 127, 61–72, doi:10.1016/j.quascirev.2015.06.001, 2015.

---

## Author Response (AR1)

October 20, 2016

Dr Bill Hu, Editor

HESSD

Dear Dr Hu,

On behalf of the authors I would like to take this opportunity to sincerely thank you and the two anonymous reviewers for the time and effort in the careful reading of our manuscript hess-2016-201, ENSO - cave dripwater hydrochemical relationship: a 7-year dataset from SE Australia. The manuscript has certainly benefited from these insightful revision suggestions.

As instructed we have now incorporated the approved changes and present you now with a final version of the manuscript for your consideration. As requested, we have also attached a marked-up manuscript version showing the changes made. We hope that this is now to your satisfaction for publication in HESS.

I look forward to hearing from you regarding our final submission.

With kind regards,

For the authors,

Carol Tadros

**Editorial review**

*I received two comments, both suggested minor revision to the manuscript. The authors have also well addressed the comments and suggestions made by the reviewers. Therefore, I suggest publish this paper if the authors well revise their manuscript according to the comments and suggestions made by the reviewers.*

We thank the Editor for this positive response. The changes have now been made to the manuscript and are detailed below. Please note, Referee comments are in blue font, italicised and underlined and the author responses are in black font. Also, page and line numbers in our responses below are references to the revised manuscript version (hess-2016-201-manuscript-version3).

**Anonymous Referee #1 comment**

*C2, General comment: "... I would recommend to always write Harrie Woods Cave in the manuscript and only write in the "Study site" section that this caves belong to a bigger cave region, which is called Yarrangobilly Caves..."*
We agree, and for the sake of clarity, in the revised manuscript, we have replaced Yarrangobilly Cave with Harrie Wood Cave throughout the manuscript where we are referring specifically to the cave [P1, L17; P3, L18; Fig. 1(A); P7, L1; P20, L17; and P22, L15].
As suggested by the referee, the exception is the study site section and P3, L30. Therefore in the revised manuscript P3, L29-30 now reads: "…Harrie Wood Cave is located within the Yarrangobilly Caves system, which includes over 250 independent limestone caves …"

When we are referring to the karst system as a whole unit [P2, L8; P3, L23-24; P4, L12] we are more specific and use "Yarrangobilly Caves system" and "Yarrangobilly Caves (BoM station 72141)" when we are referring to the Bureau of Meteorology station (72141) or data obtained from the station [P4, L14; P10, L5; P12, L2-3]; as this is the official name of the BoM station.

*C2, General comment: "... Please, also mark Harrie Woods Cave in Figure 1 instead of Yarrangobilly Caves…"*
In the revised manuscript, Harrie Wood Cave has replaced Yarrangobilly Caves in Fig. 1(A) on P6.

**MINOR TECHNICAL CORRECTIONS:**
All the below suggestions from P2, L20 to P21, L16 refer to minor text revisions and have been corrected in the revised manuscript.

*P2, L20: Please delete the "dissolution" behind "differential" and set a comma behind the second "dissolution" in that line.*

P2, L24: The "dissolution" behind "differential" has been deleted and a comma set behind the second "dissolution" in that line.

*P2, L21: Please set a comma behind the citation parentheses.*

P2, L25: A comma has been set behind the citation parentheses.

*P3, L2: Please change "Callow et al., 2014" into "Callow et al. (2014)".*

P3, L6: The reference now reads: "Callow et al. (2014)".

*P3, L6: Please change "Markowska et al., (2015)" into "Markowska et al. (2015)".*

P3, L10: The reference now reads: "Markowska et al. (2015)".

*P3, L13: Please put "New South Wales, Australia" in parentheses.*

P3, L18: "New South Wales, Australia" was placed in parentheses and for clarity we also included the city "Yarrangobilly". The line now reads: "Harrie Wood Cave (Yarrangobilly, New South Wales, Australia)."

*P3, L17-18: Please delete "In doing so, we identified proxies which respond to the present day ENSO variability." because this fits better to the results part.*

The sentence has been deleted.

*P3, L31: Please change "...located centrally within the cave at a depth of 38 m to the surface (Fig. 1B)." to ""...located centrally within the cave (Fig. 1B) at a depth of 38 m to the surface.".*

P4, L24 has been changed to read: "…located centrally within the cave (Fig. 1B) at a depth of 38 m to the surface."

*P4, L2: Please replace "paleoclimate records" with "paleoclimate studies".*

P4, L26: "studies" now replaces "records".

*P4, L15: Please replace "Markowska et al., 2015" with "Markowska et al. (2015)".*

P5, L12: The reference now reads: "Markowska et al. (2015)".

*P4, L31: Please delete "(Fig. 1C)" in this line, because the mean annual ET is not shown in this figure, but the monthly ET.*

P4, L19: "(Fig. 1C)" which appeared after "…1961-1990." has been deleted.

*P5, L1: Please set a comma in front of "respectively".*

P4, L20: a comma has been set in front of "respectively".

*P6, Figure 1C: Please replace "or" at the second y-axes with "and".*

P6, Figure 1C: "and" has replaced "or" at the second y-axes.

*P7, L3-4: Please replace "Markowska et al., 2015" with "Markowska et al. (2015)".*

P7, L3-4: The reference now reads: "Markowska et al. (2015)".

*P7, L9: Please replace "Climate record" with "Meteorological data".*

P7, L9: Sub-section 3.1 has been renamed to "Meteorological data".

*P7, L15: Please add a comma in front of "respectively".*

P7, L15: we placed a comma in front of "respectively".

*P8, L7: Please replace "...exposures above the surface…" with "...exposures above the cave".*

P8, L7: We replaced "…exposures above the surface…" with "…exposures above the cave".

*P8, L25: Please add "concentration behind "Ca", because this is no ratio.*

P8, L25: "concentration" was inserted after "Ca".

*P9, L1-2: Please change "... highest Mg content and the paleokarst samples (R13, R14) have the lowest Mg and Sr concentrations." to "... highest Mg/Ca ratio and the paleokarst samples (R13, R14) have the lowest Mg/Ca and Sr/Ca ratios." , because you gave ratios in Table 2.*

P9, L1-2: The sentence has been changed and now reads: "…highest Mg/Ca ratio and the paleokarst samples (R13, R14) have the lowest Mg/Ca and Sr/Ca ratios."

*P9, L3: Please add "concentration" behind "Ca", because this is no ratio.*

P9, L3: "concentration" was inserted after "Ca".

*P9-11: Please change Figure 2 and 3, because you first cited Figure 3, therefore it should be Figure 2.*

Changed.

Figure 2 is now on P11 and all references to Fig. 2 throughout the manuscript [P9, L10; P10, L3-4, L7 an L8; P11, L2; P12, L6; P13, L3, L16; P14, L4, L13, L26; P15, L3; P16, L21, L25; P18, L9; P19, L2, L23] have been changed.

Figure 3 is now on P12 and all references to Fig. 2 throughout the manuscript [P9, L15; P10; L2, L7, L8, L9; P12, L2; P14, L22, L27] have been changed.

*P10, L9: Please replace "Markowska et al., 2015" with "Markowska et al. (2015)".*

P10, L10: The reference now reads: "Markowska et al. (2015)".

*P12, L2: Please add "rain water" in front of d18O.*

P12, L7: "rain water" was inserted before $\delta^{18}O$.

*P12, L 5: Please change "...these recharge only events... ." to "...these recharge Events ...".*

P12, L9: "…these recharge only events... ." has been changed to read "…these recharge events …".

*P13, L7: Please add a comma in front of "respectively".*

P13, L17: we placed a comma in front of "respectively".

*P16, L5: Please replace "Sinclair et al., 2012" by "Sinclair et al. (2012).*

P16, L19: The reference now reads: "Sinclair et al. (2012)."

*P17, L25: Please replace "Markowska et al., 2015" by "Markowska et al. (2015).*

P18, L14: The reference now reads: "Markowska et al. (2015)".

*P20, Figure 6: The indication of a and b in the middle picture is switched. Please correct this.*

P21: Well spotted! This has been fixed.

*P21, L9: Please replace "McDonald et al., 2004" by "McDonald et al. (2004)".*

P22, L9: The reference now reads: "McDonald et al. (2004)".

*P21, L16: Please delete the comma in front of "(2004)".*

P22, L16: The comma has been deleted.

**CLARIFICATIONS TO TEXT**
*P1, L20: Please replace "composition" by "ratio".*
Thank you for your suggestion however we believe composition is more accurate than ratios, as variations in both concentration and ratios are being referred to here.

Thank you for drawing our attention to this oversight, we intended to write "… in response to warmer than average temperatures…" as documented in P14, L24. This has been corrected and P1, L24-25 now reads:

"…increased soil $CO_2$ production occurred in response to warmer than average temperatures…"

For clarification, the sentence "…A statistical approach was applied to classify the drip types and five flow regimes were identified …" on P3, L11-13 now reads:

"Markowska et al. (2015) applied a statistical approach to classify the drip types and identified five flow regimes which were represented using a combined conceptual flow and box hydrological model.".

We agree and in the revised manuscript Section 2 has been re-structured under section heading 'Study Area'. Study site and climate have become sub-section 2.1 (see P3, L25-26) and as suggested by the referee, "P3, L30 – P4, L3 and P4, L4-15" now appear under subheading 2.2 Drip site setting on P4, L23 – L29.

P4, L13. Median annual rainfall is correct. Long term rainfall data observations at Yarrangobilly Caves (BoM station 72141) were used to calculate the median annual rainfall, based on data between 1985-2013. The median was used as a measure of the 'average' rainfall, since extreme rainfall events bias the arithmetic mean greater than the median.

In order to remove HW2, a small adjacent calcite column that had formed from a fused stalactite-stalagmite pair approximately 10-15 cm from HW2 had to be removed also. This resulted in re-invigoration of the drip point that had formed the column, which we included in our sampling program and refer to as HW3. This information has been added to the text on P4, L26-29. In addition, supporting material has been provided to the reader in the form of a photograph and drip rate field observations of site HW2 prior to the removal of stalagmite HW2 (constituting Supplementary S1).

We agree and drip site HW3 is now represented in Fig. 1B, P6 as a filled blue circle.

Both wind speed and direction were measured (please see Supplementary S2.). This information has been added and the sentence on P7, L15-16 now reads:

"… Atmospheric measurements of pressure, humidity, … and wind speed and direction are recorded by a Davis Vantage Pro2™.".

We have provided a better explanation of this and the sentence on P9, L10-12 now reads:
"The CWB represents a residual mass curve to show the cumulative monthly water budget trends, following the method of Hurst, 1951. It is calculated as the cumulative sum of the monthly P-ET anomalies from the climatological mean (1961-1990).".

We much appreciate the reviewer's suggestions. Letter labels have been inserted on Fig. 2 (P11) and Fig. 3 (P12), improving the visual clarity for the reader.

We much appreciate the reviewer's observation. This was a typographical error and P10, L9 now reads:

" … followed by a five month period of above-average monthly rainfall from November 2011 to March 2012 (Fig. 3E)."

The solid line is the result of applying a binomial smoothing, 1 pass Gaussian filter to the monthly data. This information was inserted on P12, L3-4.

We agree. The colours for rain (Fig. 2B, P11) and rain $\delta^{18}O$ (Fig. 2C, P11) have been changed to orange and dark grey, respectively. The units for PWM are "‰" and this has been added to Fig. 2C, P11.
The scientific units for concentration of solutes in a solvent are correctly expressed as mg $l^{-1}$ (ppm or "parts per million"). As a consistent comparison with the dripwater Ca concentrations (mg $l^{-1}$) we reported weight ratios (mg $mg^{-1}$) for Mg/Ca and Sr/Ca.
In order to further improve the clarity of the axis labels on Fig. 2G, P11 and Fig. 2H, P11, the units were changed from mg $mg^{-1}$ to mg $g^{-1}$ to avoid the use of scientific notation on the Y-axis.

P13, L15 – P14, L6: Reviewer 1 refers to the Smart and Friederich (1987) classification system which characterises drip sites based on discharge response to recharge events and discharge variability. McDonald and Drysdale (2007) identified that this classification system is not applicable to southeast Australia or regions where ENSO is the main regulator of recharge and Markowska et al. (2015) showed that this classification was not appropriate for Harrie Wood Cave. As such we prefer not to classify the drip sites in this study using this classification system.

*Figure 5: The light grey circles are very hard to see. Please choose another colour for better visibility.*
P17, L1: We have darkened the grey colour for enhanced visibility.

*P20, L4: Do you mean infiltration with "Hydrological Effective Precipitation"? If this is the case infiltration is much handier.*
We agree and "Infiltration" has replaced "Hydrological Effective Precipitation" on P21, L3.

*P23, L2: Is Barbora Gallagher correct? Or should it be Barbara?*
P24, L3: This has been checked and is correct; it is the way Barbora spells her name.

**Anonymous Referee #2 comment**

*Comment: 1. You use Yarrangobilly Caves, and Harrie Wood Caves in text. Please clarify the relation between the two caves in text, and illustrate in Fig.1.*
The same comment was raised by Anonymous Referee #1 and we would like to refer you to our reply above in C2, General comment.

*Comment: 2. There are three dripwater monitoring sites (i.e. HW1, HW2, and HW3), and the distance between the three sites are less than 10m (Fig.1B). The three drip sites belong to discharge flow Type 1, mixed flow/storage connectivity. But the observation data of Ca, Mg/Ca, and Sr/Ca for HW2 are very different from HW1 and HW3. Why?*
Indeed, in the results section we document the observations that drip site HW2 is a slower dripping site (P14, L4-5) and the Ca concentrations at site HW2 are lower compared to HW1 and HW3 (P14, L15) although the Cl concentration trends are consistent (P14, L28). However, we overlooked addressing this in the discussion and are pleased the referee has pointed this out. Therefore we inserted the following interpretation as a paragraph in the discussion (P19, L7-13):

[revised manuscript text omitted]